# Chronic kidney disease biomarkers and mortality among older adults: A comparison study of survey samples in China and the United States

Hui Miao[1‡], Linxin Liu[1‡], Yeli Wang[2], Yucheng Wang[1,3], Qile He[1,4], Tazeen Hasan Jafar[5,6,7], Shenglan Tang[5], Yi Zeng[7,8], John S. Ji[1]*

1 Vanke School of Public Health, Tsinghua University, Beijing, China, 2 Health Services and Systems Research, Duke-NUS Medical School, Singapore, Singapore, 3 School of Health Humanities, Peking University, Beijing, China, 4 Institute of Medical Information, Chinese Academy of Medical Sciences and Peking Union Medical College, Beijing, China, 5 Duke Global Health Institute, Duke University, Durham, North Carolina, United States of America, 6 Department of Renal Medicine, Singapore General Hospital, Singapore, Singapore, 7 Center for the Study of Aging and Human Development, Duke University School of Medicine, Durham, North Carolina, United States of America, 8 Center for Healthy Aging and Development Studies, and Raissun Institute for Advanced Studies, Peking University, Beijing, China

‡ HM and LL are contributed equally to this work as co-first authors.
* johnji@tsinghua.edu.cn

**Data Availability Statement:** Data are available from the US CDC (https://wwwn.cdc.gov/nchs/nhanes/continuousnhanes/default.aspx) and Duke University (https://sites.duke.edu/centerforaging/

## Abstract

### Objectives

Among older adults in China and the US, we aimed to compare the biomarkers of chronic-kidney-diseases (CKD), factors associated with CKD, and the correlation between CKD and mortality.

### Setting

China and the US.

### Study design

Cross-sectional and prospective cohorts.

### Participants

We included 2019 participants aged 65 and above from the Chinese Longitudinal Healthy Longevity Study (CLHLS) in 2012, and 2177 from US National Health and Nutrition Examination Survey (NHANES) in 2011–2014.

### Outcomes

Urinary albumin, urinary creatinine, albumin creatinine ratio (ACR), serum creatinine, blood urea nitrogen, plasma albumin, uric acid, and estimated glomerular filtration rate (eGFR). CKD (ACR $\geq$ 30 mg/g or eGFR< 60 ml/min/1.73m$^2$) and mortality.

programs/chinese-longitudinal-healthy-longevity-survey-clhls/) or Peking University (https://opendata.pku.edu.cn/dataset.xhtml?persistentId=doi:10.18170/DVN/UWS2LR) for researchers who meet the criteria for access to confidential data.

**Funding:** The Chinese Longitudinal Healthy Longevity Study (CLHLS) datasets analyzed in this paper are jointly supported by the National Key R&D Program of China (2018YFC2000400), National Natural Sciences Foundation of China (72061137004), Duke/Duke-NUS/RECA(Pilot)/2019/0051, and the U.S. National Institute of Aging of National Institute of Health (P01AG031719). Collections of the Chinese Longitudinal Healthy Longevity Surveys (CLHLS) datasets analyzed in this paper were jointly supported by the National Key R&D Program of China (2018YFC2000400), National Natural Sciences Foundation of China (72061137004), and the U.S. National Institute of Aging/ National Institute of Health (P01AG031719).

**Competing interests:** The authors have declared that no competing interests exist.

## Analytical approach

Logistic regression and Cox proportional hazard models. Covariates included age, sex, race, education, income, marital status, health condition, smoking and drinking status, physical activity and body mass index.

## Results

Chinese participants had lower levels of urinary albumin, ACR, and uric acid than the US (mean: 25.0 vs 76.4 mg/L, 41.7 vs 85.0 mg/g, 292.9 vs 341.3 μmol/L). In the fully-adjusted model, CKD was associated with the risk of mortality only in the US group (hazard ratio [HR], 95% CI: 2.179, 1.561–3.041 in NHANES, 1.091, 0.940–1.266 in CLHLS). Compared to eGFR≥90, eGFR ranged 30–44 ml/min/1.73m$^2$ was only associated with mortality in the US population (HR, 95% CI: 2.249, 1.141–4.430), but not in the Chinese population (HR, 95% CI: 1.408, 0.884–2.241).

## Conclusions

The elderly participants in the US sample had worse CKD-related biomarker levels than in China sample, and the association between CKD and mortality was also stronger among the US older adults. This may be due to the biological differences, or co-morbid conditions.

## Introduction

Chronic kidney disease (CKD) is a public health problem around the world. It refers to kidney damage or lasting low glomerular filtration rate (GFR), regardless of causes [1]. CKD is associated with premature cardiovascular disease, and other complications of progressive CKD including anemia, bone disease, and end stage kidney disease. In high-income and middle-income countries, about one in ten people have CKD [2]. Globally, the all-age prevalence of CKD has increased by 29.3% since 1990 [3]. In 2017, 1.2 million people around the world died from CKD, and the global all-age mortality rate from CKD increased by 41.5% between 1990 and 2017 [3]. CKD lowers the quality of life and leads to catastrophic health expenditures [2]. Therefore, the prevalence and trend of CKD is a worldwide public health threat [1].

Previous studies revealed substantial difference between China and USA in the prevalence and mortality of CKD. In 2019, the estimated age-standardized prevalence of CKD in China and the US were 8125 and 8179 per 100,000, respectively. The percentage of changes in age-standardized prevalence rates between 2010 and 2019 of the two countries were 0.6% and 4% respectively. Age-standardized mortality of CKD in China in 2019 was 11.2 per 100 000 (95% CI: 9.6, 12.8), lower than that in the US (17.8 per 100 000 [95% CI: 16.1, 18.9]), and the age-standardized mortality change between 2010 and 2019 were also different (-10.1% [95% CI: -23.1, 3.1] vs. 7.1% [95% CI: 3.4, 11.3]) [4]. Previous studies found that people in China and the US share many common risk factors for CKD; however, differences were observed for the prevalences of these risk factors as well as their potential impact on CKD [5, 6]. The Chinese population had a lower prevalence of adjusted albuminuria, decreased eGFR and CKD than the US, and 65% of the difference of decreased eGFR between the countries was explained by risk factors (diabetes, hypertension, CVD, hyperuricemia, and obesity) together with age and sex [5, 6]. In a cross-sectional study, the prevalence of stage 3 and stage 4 chronic kidney disease in China was lower compared with those in the US (stage 3 [95% CI]: 1.6% [1.4–1.8] vs.

7.69% [7.02–8.38], stage 4: 0.1% [0.06–0.2] vs. 0.35% [0.25–0.45]) [7, 8]. However, these previous studies mainly focused on comparing the prevalence of CKD in the general population between China and the US, the differences in the prevalence of CKD among elderly populations between China and the US, and their association with mortality was not known yet.

Therefore, we aimed to compare the prevalence and mortality risk of CKD among elderly people between China and the US. We hypothesized that the elderly in China had a lower prevalence of CKD and a lower mortality rate compared to the US counterparts. In addition, the association between CKD and mortality might also be different among the US elderly population and the Chinese.

## Methods

### Study population

We used data from the Chinese Longitudinal Healthy Longevity Study (CLHLS) and National Health and Nutrition Examination Survey (NHANES) to compare the CKD-related biomarkers, CKD, and mortality among the older population, aged 65 years or older. Both CLHLS and NHANES collected data through in-person interviews, and blood and urine samples. All procedures in the NHANES survey cycles used in this study were approved by the National Center for Health Statistics Research Ethics Review Board (Protocol #2011–17), and written informed consent was obtained from all participants. The CLHLS study was approved by research ethics committees of Duke University and Peking University (IRB00001052-13074) and written informed consent was obtained from each respondent. The brief introduction of the two cohorts we used could be found in S1 Table.

The Chinese participants were sampled from the Healthy Ageing and Biomarkers Cohort Study (HABCS), which is a sub-cohort of CLHLS. The study surveyed eight longevity area (Laizhou City of Shandong Province, Xiayi County of Henan Province, Zhongxiang City of Hubei Province, Mayang County of Hunan Province, Yongfu County of Guangxi Autonomous Area, Sanshui District of Guangdong Province, Chengmai County of Hainan Province and Rudong County of Jiangsu Province), covering various geographical and climatic regions of China. The study design and sampling method were described previously [9]. For the current analysis, we used data from the 2011–2012 wave. Between May and September 2012, 2354 participants aged 65 years and older received the face-to-face interview and took the blood and urine test. We excluded 94 participants without urine sample test (Urinary albumin and urine creatinine) and blood sample test (serum creatinine, BUN and plasma albumin), 214 participants with blood test but without urine test, and 28 participants with urine test but without blood test. We finally included 2019 participants with test results of the urinary albumin, urinary creatinine, serum creatinine, ACR, BUN, plasm albumin, and uric acid for the description and risk factor analysis. The CLHLS participants were followed up in 2014 and 2017/2018. We further excluded 321 participants lost in the first follow-up and included 1698 participants for the mortality analysis. In CLHLS, those who did not have the study biomarker result were older, more likely to be female, without formal education and widowed. Those lost in the first follow-up were more likely to have formal education and higher household income, less likely to do physical activity currently than those included in the follow-up study. Meanwhile, they had similar age, gender, race, marriage, smoking, alcohol drinking and BMI distribution (S2–1 Table).

The US participants were from the National Health and Nutrition Examination Survey, which is a continuous, nationally representative survey of the non-institutionalized US population of all ages [10]. NHANES uses complex sampling design recruiting people, and included a subgroup of ethnic minorities. Data from the 2011–2012 and 2013–2014 waves were used for

the current analysis. Among 2556 participants aged 65 and over, we excluded 379 subjects with missing urinary albumin, creatinine and albumin creatinine ratio (ACR), blood urea nitrogen (BUN), serum creatinine, uric acid, or plasma albumin. In NHANES, those with missing biomarker data were slightly older, more likely to be female, widowed, with lower education level and income (S2–2 Table). We eventually included 2177 participants for the current study. The participants were followed up in 2015 when 201 of them were found dead.

Participants in CLHLS were interviewed and taken urine and blood test between May and September 2012, were followed up in 2014 and 2017/2018. The recruitment and involvement of participants were organized by Center for Healthy Aging and Development Studies in Peking University National School of Development. Participants in NHANES were interviewed and examined by the US Centers for Disease Control and Prevention. The details of participants' involvement in NHANES and CLHLS were introduced in the survey documentation of NHANES [11] and CLHLS [12].

## Measurement of biomarkers

In CLHLS, the urine was tested for albumin and creatinine using Siemens Microalbustix (Siemen Healthcare Diagnostic, USA). Blood plasma analyses were determined by an Automatic Biochemistry Analyzer (Hitachi 7180, Japan) using commercially available diagnostic kits (Roche Diagnostic, Mannheim, Germany), and serum creatinine was determined by the picric acid method, BUN was determined by urease ultraviolet rate method, and blood uric acid was determined by uricase colorimetric method. The central clinical laboratory at Capital Medical University conducted all laboratory analyses in Beijing. Estimated glomerular filtration rate (eGFR) was calculated by the original Chronic Kidney Disease Epidemiology Collaboration (CKD-EPI) creatinine-based equation that was validated among the Chinese population (S3 Table) [13, 14].

In NHANES, urinary albumin was measured by a solid-phase fluorescent Immunoassay described by Chavers et al. [15]. In 2011–2012, urinary creatinine was measured on the Roche/Hitachi Mod P chemistry analyzer; while in 2013–2014, it was measured on the Roche/Hitachi Cobas 6000 chemistry analyzer. Urine specimens were analyzed in the University of Minnesota, Minneapolis, MN. The urine albumin/creatinine ratio was created, with the random urine albumin in ug/mL divided by urine creatinine in mg/dL, and then multiplying 100, round to 0.01. DxC800 determined the concentration of serum creatinine, BUN, plasma albumin and uric acid by means of the Jaffe rate method, the enzymatic conductivity rate method, a bichromatic digital endpoint method and a timed endpoint method, respectively. Serum specimens were shipped to the Collaborative Laboratory Services, Ottumwa, Iowa for analysis. Estimated GFR was calculated using the original CKD-EPI creatinine-based equation as in the CLHLS part (S3 Table).

## Outcomes

CKD was defined as: ACR $\geq$ 30 mg/g or eGFR$<$ 60 ml/min/1.73m$^2$ according to the "KDIGO clinical practice guidelines for chronic kidney disease: Evaluation, classification, and stratification" [16].

In CLHLS, the immediate family members of subjects reported the mortality information in the follow-up surveys in 2014 and 2017/2018. The survival time was entered as month counted from the month of the initial interview to the month of death or the last interview time. NHANES is linked to the National Death Index (NDI) and other data files which enable us to identify mortality. The mortality follow-up time was calculated using person-months from the mobile examination center date to the date of death. For those alive, the follow-up

time was calculated using the end of the mortality period (December 31, 2015) [17]. Among the 2019 participants included in the CLHLS part, 321 lost in the first follow-up in 2014. In the NHANES part, only 3 participants did not have available mortality data.

## Covariates

We selected some available demographics, social-economic status, lifestyle and health indicators in both CLHLS and NHANES: age, sex, race/ethnicities, education, household income, marital status, health condition, smoking status, drinking status, physical activity, and body mass index (BMI), hypertension and diabetes as covariates. In CLHLS, the age was calculated as the difference between the interview dates and birth dates, and verified by the investigator. In NHANES, age was obtained at the time of screening, and participants age 80 and over were coded as 80, which led to a younger average age. In CLHLS, the ethnicity was coded as Han Chinese and ethnic minorities; while in NHANES, it was categorized into Mexican American, other Hispanic, non-Hispanic White, non-Hispanic Black, non-Hispanic Asian, and other races. For participants from CLHLS, we defined having at least one year's schooling as formal education, and the others as no formal education. The educational degree in NHANES was classified as below high school, high school, and college or above. In CLHLS, annual household income of one year before the interview year was recorded and categorized into tertiles. However, family income in NHANES was indicated by the ratio of family income to poverty (PIR), which is the ratio of total family income to the poverty threshold for the year of the interview, divided into low income (0–1.85), middle income (1.86–3.50), and high income (>3.51). For the marital status, both CLHLS and NHANES included married, separated, divorced, widowed, and never married, but NHANES had an additional category of "living with partner". The self-reported health condition included very good, good, fair, bad, and very bad in CLHLS, but were coded as excellent, very good, good, fair, and poor in NHANES. According to the CLHLS participants' answer to the questions "do you currently smoke/drink alcohol/exercise regularly?" and "did you smoke/drink alcohol/exercise regularly in the past?", we coded the smoking and alcohol drinking status as "Current", "Former", and "Never", and coded the currently exercise variable as yes or no. In NHANES, we coded smoking and alcohol drinking behavior as "Current", "Former", and "Never" bases on the questionnaires, and having physical activity was defined as taking vigorous or moderate work/recreational activities, or walking or using bicycle in a typical week. We calculated BMI as body weight divided by the square of the body height (unit: $kg/m^2$). We used WHO standard of BMI in both CLHLS and NHANES, which defined a BMI of $<18.5$ $kg/m^2$ as underweight, a BMI of $\geq18.5$ to $<25$ $kg/m^2$ as normal weight, a BMI of $\geq25$ to $<30$ $kg/m^2$ as overweight, and a BMI of $\geq30$ $kg/m^2$ as obese. We defined hypertension as systolic blood pressure $\geq140$mmHg and/or diastolic blood pressure $\geq90$mmHg, and used self-reported diabetes.

## Statistical analysis

We first summarized the demographic characteristics and biomarkers level and then performed logistic regression to analyze the risk factors associated with CKD. CKD was defined as: ACR $\geq$ 30 mg/g or eGFR< 60 ml/min/1.73m². The odds ratio (OR) and 95% confidence interval (CI) were reported. We conducted Cox proportional hazard model to evaluate the individual association of CKD-related biomarkers, eGFR levels, and CKD status with mortality. The Cox models adjusted for age, sex, race/ethnicities, education, household income, marital status, health condition, smoking status, drinking status, physical activity, and BMI. Missing value of covariates were coded as a categorical variable and included in the logistic regression and the Cox models. Besides, age was adjusted for as four separate age bins (65–69,

70–74, 75–79, 80+) in the logistic models and as continuous in the Cox models. We further ran the logistic and cox models for different age groups. We calculated hazard ratios (HR) and 95% CI to reflect the effect size. We performed all the analyses both to the Chinese and American samples. We presented results considering sample weight in S7–1–S9 Tables. We used R 3.6.1 and SAS university edition for the analysis.

### Data sharing statement

The NHANES data that support the findings of this study are openly available at https:// wwwn.cdc.gov/nchs/nhanes/continuousnhanes/default.aspx while the CLHLS data are available on request at https://sites.duke.edu/centerforaging/programs/chinese-longitudinal-healthy-longevity-survey-clhls/data-downloads/.

### Patient and public involvement

There was no patient or public involvement in this study.

## Results

### Demographic characteristics

The proportion of older adults over 80 in CLHLS was 65.5% with a mean age of 85.7 (SD: 12.2), larger than 26.1% in NHANES with a mean age of 73.2 (SD: 5.4). Educational attainment was drastically different, in CLHLS, most of the participants had no formal education (61.3%), while in NHANES, about 70% of the participants received high school education or above. Also, more Chinese participants were widowed (56%) than US participants (25.8%). In CLHLS, more participants rated their health condition as "Good" than in NHANES (37.1% "Good" versus 23.7% "Very good", "Fair" in CLHLS and "Very good" in NHANES were the top level of self-reported health status), while fewer Chinese participants rated as "Bad" than US participants (10% "Bad" versus 23.4% "Fair", both were the fourth level of self-reported health condition). More Chinese participants reported that they were never smoker (72.6%) than US participants (50.3%), and more Chinese participants never drank any alcohol (75.7%) than US participants (18.0%). Besides, 39.9% of the US participants had physical activities, more than that of Chinese participants (15.4%). CLHLS had much more underweight participants (23.6%) than NHANES (1.7%), while NHANES had much more overweight and obese participants (35.4% and 34.1%, respectively) than CLHLS (10.6% and 3.1%, respectively). There were also more participants had hypertension and less participants had self-reported diabetes in CLHLS than NHANES (56.6% vs. 34.3% and 2.4% vs. 24.2% respectively) (Tables 1 and 2).

### Prevalence of CKD in both countries

Among people over 65 years of age, the prevalence of CKD in CLHLS and NHANES samples were 44.4% (95% CI: 42.2%-46.6%) and 42.3% (95% CI: 40.2%-44.6%), respectively. Besides, the CKD prevalence for the participants without diabetes and hypertension were 37.8% in CLHLS and 33% in NHANES (diabetes/hypertension prevalence: 58% in CLHLS vs. 50% in NHANES). However, the prevalence in each age group of NHANES participants was higher than Chinese participants, as 27.4% vs 17.9% for those of 65–69 years old, 35.6% vs 18.3% for those aged 70–74, 49.6% vs 31.8% for 75–79 years old, and 62.3% vs 56.0% for the eldest group. In both CLHLS and NHANES, people who were aged over 80 (56.0% vs 62.3%) or widowed (54.6% vs 52.0%), or had lower educational level (51.3% of "no formal education" vs 49.0% of "below high school") or bad health condition (52.2% of "bad" vs 51.9% of "fair") usually had a

**Table 1. Demographic characteristics and mean (SD) of biomarkers (Chinese participants: CLHLS 2012).**

| Characteristics | n (%) | CKD n (%) | Urinary albumin (mg/L) | Urinary creatinine (mg/dL) | Albumin creatinine ratio (mg/g) | Serum creatinine (μmol/L) | Blood urea nitrogen (mmol/L) | Plasma albumin (g/L) | Uric acid (μmol/L) | eGFR (mL/min per 1.73 m²) |
|---|---|---|---|---|---|---|---|---|---|---|
| **Total** | 2019 (100) | 896 (44.4) | 25 (75.4) | 106.6 (68.3) | 41.7 (231.5) | 82.1 (29.2) | 6.9 (2.1) | 40.3 (4.9) | 292.9 (90.4) | 67.5 (18.1) |
| **Range [min, max]** | / | / | [0, 991.7] | [0.02, 479.2] | [0, 6417.9] | [27, 464] | [2.3, 24.8] | [18.5, 57.1] | [0.7, 935.8] | [8.9, 116.1] |
| **Age (mean ± SD)** | 85.7 ±12.2 | 90.5 ±10.8 | / | / | / | / | / | / | / | / |
| **Age group** | | | | | | | | | | |
| 65–69 | 240 (11.9) | 43 (17.9) | 16.9 (59.8) | 125.5 (68.4) | 17.6 (61.3) | 75.4 (20.1) | 6.3 (1.6) | 42.3 (4.0) | 286.2 (93.4) | 84.1 (14.5) |
| 70–74 | 240 (11.9) | 44 (18.3) | 10.2 (20.3) | 114.1 (67.7) | 11.5 (26.0) | 77.8 (18.9) | 6.4 (1.7) | 42.2 (4.8) | 287.2 (80.5) | 79.2 (13.8) |
| 75–79 | 217 (10.7) | 69 (31.8) | 23 (82.3) | 121.3 (71.4) | 19.9 (63.4) | 82.8 (25.9) | 6.5 (1.9) | 41.2 (4.5) | 293.9 (99.4) | 72.8 (16.0) |
| 80+ | 1322 (65.5) | 740 (56.0) | 29.5 (82.4) | 99.4 (66.8) | 55.2 (282.7) | 84 (32.2) | 7.1 (2.3) | 39.4 (5.0) | 294.9 (90.0) | 61.5 (16.5) |
| **Gender** | | | | | | | | | | |
| Male | 933 (46.2) | 307 (32.9) | 24.1 (83.2) | 125.3 (70.1) | 25.2 (99.7) | 89.3 (31.1) | 6.9 (2.1) | 40.7 (4.8) | 314.8 (90.5) | 71.8 (17.6) |
| Female | 1086 (53.8) | 589 (54.2) | 25.8 (68.0) | 90.6 (62.5) | 56 (301.3) | 75.9 (25.8) | 6.8 (2.2) | 39.9 (5.0) | 274 (86.0) | 63.8 (17.8) |
| **Race** | | | | | | | | | | |
| Han Chinese | 1817 (90.0) | 799 (44.0) | 23.5 (70.8) | 104.1 (66.3) | 41.7 (238.4) | 81.5 (27.9) | 6.8 (2.1) | 40.5 (4.9) | 291.5 (89.4) | 67.8 (18.0) |
| Ethnic minorities | 152 (7.5) | 70 (46.1) | 37.3 (110.5) | 137.3 (78.2) | 38.5 (168.3) | 90.2 (41.6) | 7.1 (2.7) | 38.1 (4.9) | 315.2 (96.8) | 64.2 (19.3) |
| Missing | 50 (2.5) | 27 (54.0) | 41.7 (100.7) | 106.6 (85.2) | 50.3 (113.5) | 79.2 (22.9) | 6.8 (1.7) | 39.4 (4.7) | 272.9 (97.9) | 66 (17.8) |
| **Education** | | | | | | | | | | |
| No formal education | 1238 (61.3) | 635 (51.3) | 27.2 (78.5) | 97.5 (65.6) | 52.3 (284.9) | 79.6 (29.5) | 6.9 (2.2) | 39.7 (4.8) | 283.3 (90.0) | 64.6 (17.7) |
| Formal education | 764 (37.8) | 250 (32.7) | 21.1 (70.1) | 121.6 (70.2) | 23.6 (95.6) | 86.1 (28.4) | 6.8 (2.1) | 41.2 (5.1) | 308.6 (89.2) | 72.4 (17.9) |
| Missing | 17 (0.8) | 11 (64.7) | 41.4 (69.1) | 97 (60.5) | 87.5 (154.5) | 82.9 (19.0) | 7.2 (2.0) | 38 (5.8) | 282.7 (78.3) | 62.7 (16.1) |
| **Household income (RMB)** | | | | | | | | | | |
| Tertile 1 (<6,000) | 637 (31.6) | 235 (36.9) | 21.3 (72.6) | 106 (67.2) | 37.1 (272.9) | 77.9 (23.4) | 6.8 (1.9) | 40 (4.5) | 272.4 (83.5) | 70.8 (17.3) |
| Tertile 2 (6,000–19,000) | 661 (32.7) | 291 (44.0) | 24.7 (74.3) | 108.6 (67.5) | 42.2 (223.1) | 81 (25.2) | 6.9 (2.1) | 40.3 (4.8) | 295.6 (87.7) | 67.9 (17.5) |
| Tertile 3 (20,000-more than 100,000) | 572 (28.3) | 288 (50.3) | 29.8 (85.3) | 108.3 (73.7) | 47.2 (216.6) | 87.5 (36.8) | 6.9 (2.4) | 40.7 (5.5) | 308.8 (87.3) | 64.5 (18.7) |
| Missing | 149 (7.4) | 82 (55.0) | 23.8 (45.4) | 94.4 (53.1) | 38.4 (86.3) | 84.2 (31.5) | 6.9 (2.2) | 39.6 (5.2) | 307.2 (121.3) | 63.2 (19.3) |
| **Marital Status** | | | | | | | | | | |
| Married | 774 (38.3) | 229 (29.6) | 20.2 (74.4) | 120.8 (69.5) | 19.2 (60.4) | 83 (28.6) | 6.6 (1.9) | 41.5 (4.6) | 299.6 (92.0) | 74.5 (17.0) |
| Separated | 40 (2.0) | 18 (45.0) | 22.6 (54.9) | 117.1 (79.8) | 16.3 (33.6) | 87.3 (22.9) | 6.5 (2.0) | 38.8 (4.5) | 302.5 (72.9) | 68.8 (16.7) |

(*Continued*)

**Table 1.** (Continued)

| Characteristics | n (%) | CKD n (%) | Urinary albumin (mg/L) | Urinary creatinine (mg/dL) | Albumin creatinine ratio (mg/g) | Serum creatinine (μmol/L) | Blood urea nitrogen (mmol/L) | Plasma albumin (g/L) | Uric acid (μmol/L) | eGFR (mL/min per 1.73 m$^2$) |
|---|---|---|---|---|---|---|---|---|---|---|
| Divorced | 5 (0.2) | 1 (20.0) | 10.8 (19.0) | 124.8 (113) | 15.2 (28.7) | 70.2 (20.2) | 6.7 (1.5) | 40 (3.5) | 257.7 (74.3) | 88.5 (16.7) |
| Widowed | 1131 (56.0) | 618 (54.6) | 28.6 (78.4) | 96.4 (65.5) | 59 (303.8) | 81.3 (29.8) | 7 (2.3) | 39.4 (5.0) | 286 (86.5) | 62.5 (17.2) |
| Never married | 20 (1.0) | 6 (30.0) | 16.4 (36.8) | 114.6 (52.5) | 13.9 (32.8) | 83.7 (25.5) | 6.6 (1.3) | 40.5 (5.7) | 289.4 (113.2) | 77.2 (21.3) |
| Missing | 49 (2.4) | 24 (49.0) | 24.8 (38.9) | 104 (64.4) | 34.8 (72.9) | 82.4 (30.6) | 6.7 (2.4) | 40.8 (5.2) | 340.5 (130.4) | 64.3 (18.5) |
| **Health condition** | | | | | | | | | | |
| Very good | 103 (5.1) | 41 (39.8) | 15.7 (29.8) | 116.5 (68.9) | 20 (48.8) | 82.2 (19.6) | 6.7 (1.9) | 40.8 (4.7) | 306.3 (86.0) | 68.9 (16.5) |
| Good | 750 (37.1) | 286 (38.1) | 25.7 (85.5) | 109.6 (68.6) | 47.3 (307.9) | 82.9 (33.4) | 7 (2.3) | 40.5 (4.9) | 292.6 (92.9) | 69.3 (18.8) |
| Fair | 775 (38.4) | 367 (47.4) | 24.7 (66.8) | 106.3 (69.4) | 40.3 (200.1) | 82.2 (26.3) | 6.8 (2.0) | 40.6 (4.7) | 291.8 (83.9) | 66.9 (17.9) |
| Bad | 201 (10.0) | 105 (52.2) | 24 (77.6) | 103.8 (68.3) | 29.3 (76.8) | 84.2 (28.9) | 6.6 (2.2) | 39.5 (5.0) | 299.6 (94.7) | 65 (18.3) |
| Very Bad | 12 (0.6) | 4 (33.3) | 13.7 (21.8) | 79.5 (47.6) | 20.9 (34.1) | 68.4 (24.6) | 6.2 (1.9) | 37.5 (8.2) | 250 (52.1) | 73 (16.3) |
| Missing | 178 (8.8) | 93 (52.2) | 30.7 (83.2) | 95 (61.9) | 52.3 (160.3) | 76.8 (26.8) | 7.1 (2.2) | 38.3 (5.5) | 286 (104.6) | 64.1 (16.2) |
| **Smoking status** | | | | | | | | | | |
| Never smoker | 1465 (72.6) | 700 (47.8) | 26.6 (80.7) | 100.9 (66.2) | 48.5 (268.7) | 80.5 (29.9) | 6.9 (2.2) | 40.2 (5.0) | 286.1 (87.8) | 66.3 (18.0) |
| Former smoker | 164 (8.1) | 60 (36.6) | 18 (40.6) | 119.2 (74.6) | 27.1 (67.0) | 85.6 (23.2) | 7 (2.1) | 39.7 (4.7) | 304.5 (82.6) | 69 (16.2) |
| Current smoker | 334 (16.5) | 108 (32.3) | 20.8 (67.9) | 127.2 (71.8) | 19.6 (57.1) | 87.2 (27.7) | 6.5 (1.9) | 40.4 (4.9) | 309.2 (93.2) | 72.6 (18.7) |
| Missing | 56 (2.8) | 28 (50.0) | 27.9 (43.4) | 97.3 (54.6) | 38.9 (75.5) | 82 (29.4) | 6.9 (2.4) | 41.7 (4.9) | 338.6 (129.2) | 65.3 (18.3) |
| **Drinking status** | | | | | | | | | | |
| Never drinker | 1528 (75.7) | 708 (46.3) | 25.8 (78.3) | 103.7 (67.6) | 45.8 (258) | 81.5 (30.2) | 6.9 (2.1) | 40.2 (4.9) | 285.7 (87.7) | 66.5 (18.1) |
| Former drinker | 120 (5.9) | 54 (45) | 34.7 (109.5) | 116.4 (71.4) | 40.6 (185.7) | 87.6 (28.0) | 7 (2.3) | 39.3 (5) | 313.3 (93.6) | 67.7 (18.0) |
| Current drinker | 315 (15.6) | 105 (33.3) | 18 (43.4) | 116.6 (70.1) | 24.2 (80.6) | 82.8 (23.9) | 6.5 (1.9) | 40.9 (4.8) | 310.7 (88.9) | 72.7 (17.4) |
| Missing | 56 (2.8) | 29 (51.8) | 22.5 (37.1) | 109.4 (67.3) | 31 (68.9) | 83.1 (28.9) | 6.9 (2.4) | 40.5 (5.7) | 343 (124.3) | 64.9 (18.4) |
| **Physical activity** | | | | | | | | | | |
| Yes | 311 (15.4) | 144 (46.3) | 21.6 (58.1) | 121.7 (81.0) | 47.7 (368.8) | 85.6 (23.4) | 6.8 (2.0) | 41.1 (4.8) | 309.2 (86.6) | 67.1 (17.4) |
| No | 1598 (79.1) | 701 (43.9) | 25.9 (80.3) | 103.2 (65.3) | 41.5 (202.7) | 81.3 (29.4) | 6.9 (2.1) | 40.1 (4.9) | 287.4 (86.9) | 67.6 (18.2) |
| Missing | 110 (5.4) | 51 (46.4) | 22.3 (35.1) | 113.3 (67.4) | 28.1 (55.4) | 82.9 (38.3) | 6.6 (2.4) | 40.4 (5.5) | 325.6 (130.9) | 67.2 (19.5) |
| **Body mass index (kg/m$^2$)** | | | | | | | | | | |
| Underweight (<18.5) | 477 (23.6) | 271 (56.8) | 25.5 (71.3) | 98.1 (61.2) | 58.4 (377.9) | 83.1 (28.1) | 6.9 (2.2) | 39.2 (5.0) | 286 (85.0) | 62.2 (18.0) |

(*Continued*)

**Table 1.** (Continued)

| Characteristics | n (%) | CKD n (%) | Urinary albumin (mg/L) | Urinary creatinine (mg/dL) | Albumin creatinine ratio (mg/g) | Serum creatinine (µmol/L) | Blood urea nitrogen (mmol/L) | Plasma albumin (g/L) | Uric acid (µmol/L) | eGFR (mL/min per 1.73 m²) |
|---|---|---|---|---|---|---|---|---|---|---|
| Normal (18.5–24.9) | 1153 (57.1) | 469 (40.7) | 23.4 (75.3) | 108.2 (69.4) | 36.3 (168.8) | 81.9 (29.4) | 6.8 (2.1) | 40.4 (4.7) | 290.6 (91.4) | 69.1 (17.7) |
| Overweight (25.0–29.9) | 229 (11.3) | 73 (31.9) | 23.1 (66.2) | 123.1 (75.4) | 24.1 (71.1) | 81.4 (27.2) | 6.8 (1.9) | 42.2 (5.0) | 312.1 (90.2) | 72.9 (17.9) |
| Obese (> = 30) | 58 (2.9) | 23 (39.7) | 41 (112.8) | 106.3 (69.7) | 66 (262.4) | 83.4 (38.1) | 7.1 (3.1) | 41.4 (5.5) | 307.6 (83.3) | 68.4 (18.8) |
| Missing | 102 (5.1) | 60 (58.8) | 36 (87.0) | 91.9 (63.5) | 50.4 (138.4) | 80.4 (30.1) | 6.7 (2.4) | 38.2 (5.1) | 299.5 (102.4) | 61.7 (17.6) |
| **Hypertension** | | | | | | | | | | |
| Yes | 1142 (56.6) | 558 (48.9) | 28.9 (81.5) | 100.3 (66.9) | 56.4 (300.2) | 82.5 (29.8) | 6.8 (2.2) | 40.7 (4.8) | 296.5 (87.1) | 66.3 (17.9) |
| No | 857 (42.4) | 326 (38.0) | 19.3 (64.1) | 114.2 (68.4) | 22.4 (73.8) | 81.4 (28.3) | 6.9 (2.1) | 39.8 (5.0) | 287.6 (93.9) | 69.3 (18.3) |
| Missing | 20 (1.0) | 12 (60.0) | 47.6 (131.9) | 147 (97.2) | 30.1 (72.6) | 87.6 (27.8) | 7.3 (1.8) | 35.8 (4.3) | 310.4 (116.6) | 63.3 (17.9) |
| **Diabetes** | | | | | | | | | | |
| Yes | 48 (2.4) | 22 (45.8) | 46.5 (128.8) | 111.1 (67.6) | 41.5 (101.3) | 84 (32.5) | 6.6 (1.6) | 41 (5.2) | 303.4 (97.4) | 71.1 (19.3) |
| No | 1940 (96.1) | 856 (44.1) | 24.2 (73.6) | 106.5 (68.4) | 41.7 (235.5) | 82 (29.0) | 6.9 (2.1) | 40.3 (4.9) | 292.6 (90.3) | 67.6 (18.1) |
| Missing | 31 (1.5) | 18 (58.1) | 41.2 (72.8) | 110.2 (67.7) | 43.3 (63.9) | 86.2 (33.2) | 7.5 (2.8) | 37.9 (6.3) | 290.3 (85.1) | 58.1 (17.8) |

Abbreviations: SD = standard deviation, CKD = chronic kidney diseases, eGFR = estimated glomerular filtration rate, RMB = renminbi.

higher prevalence of CKD. Besides, Chinese participants who were female (54.2%) or underweight (56.8%), or had the highest household income (50.3%), never smoked (47.8%), never drank (46.3%), did physical activity (46.3%), or had hypertension (48.9%) tended to have higher CKD prevalence. However, in the US, the prevalence was higher among people who were non-Hispanic White (44.9%), former smoker (44.2%), former drinker (48.1%), obese (44.9%), had income (PIR) <1.87 (44.8%), did not do physical activity (45.6%), had hypertension (49.7%) or self-reported diabetes (57.4%).

## Biomarkers' level by the demographic characteristics

The mean values of urinary albumin (mg/L), urinary creatinine (mg/dL), ACR (mg/g), serum creatinine (µmol/L), plasma albumin (g/L), uric acid (µmol/L) and eGFR (mL/min per 1.73 m²) were lower in the Chinese participants than the US sample (25.0 vs 76.4, 106.6 vs 109.4, 41.7 vs 85.0, 82.1 vs 92.0, 40.3 vs 41.8, 292.9 vs 341.3, and 67.5 vs 69.7), while the BUN (mmol/L) was higher in the Chinese sample (6.9 vs 6.1) (Tables 1 and 2). The median comparison showed the same trend (S4–1 and S4–2 Table).

The mean difference between the CLHLS and NHANES varied in different age groups. The urinary creatinine, BUN and eGFR were both higher in CLHLS participants than NHANES sample aged younger than 80 and this difference almost disappeared in participants aged 80 and older. The eGFR decreased with age increasing in both CLHLS and NHANES samples. The urinary albumin, ACR, serum creatinine, and uric acid were lower in Chinese participants than the US ones among different age groups consistently. Plasma albumin level was similar in both samples across all age groups.

**Table 2. Demographic characteristics and mean (SD) of biomarkers (US participants: NHANES 2011–2014).**

| Characteristics | n (%) | CKD n (%) | Urinary albumin (mg/L) | Urinary creatinine (mg/dL) | Albumin creatinine ratio (mg/g) | Serum creatinine (μmol/L) | Blood urea nitrogen (mmol/L) | Plasma albumin (g/L) | Uric acid (umol/L) | eGFR (mL/min per 1.73 m²) |
|---|---|---|---|---|---|---|---|---|---|---|
| **Total** | 2177 (100) | 921 (42.3) | 76.4 (359.6) | 109.4 (68.8) | 85.0 (449.3) | 92.0 (42.5) | 6.1 (2.8) | 41.8 (3.1) | 341.3 (88.0) | 69.7 (19.3) |
| **Range [min, max]** | / | / | [0.2, 7410.0] | [9.0, 567.0] | [0.3, 10465.1] | [35.4, 818.6] | [1.4, 33.9] | [21.0, 52.0] | [65.4, 701.9] | [5.7, 114.8] |
| **Age (mean ± SD)** | 73.2 ±5.4 | 75.0 ±5.1 | / | / | / | / | / | / | / | |
| **Age group** | | | | | | | | | | |
| 65–69 | 682 (31.3) | 187 (27.4) | 61.2 (297.5) | 115.7 (76.3) | 71.0 (494.1) | 87.0 (43.1) | 5.4 (2.6) | 42.2 (3.0) | 335.0 (86.3) | 77.9 (18.3) |
| 70–74 | 567 (26.1) | 202 (35.6) | 52.0 (218.5) | 106.1 (63.6) | 47.1 (180.5) | 86.4 (31.7) | 5.7 (2.1) | 42.0 (3.0) | 338.1 (81.8) | 72.4 (16.8) |
| 75–79 | 361 (16.6) | 179 (49.6) | 109.3 (484.7) | 112.0 (72.5) | 116.4 (524.2) | 98.7 (53.9) | 6.4 (2.9) | 41.6 (3.1) | 352.1 (95.9) | 64.9 (19.0) |
| 80+ | 567 (26.1) | 353 (62.3) | 98.0 (437.7) | 103.5 (60.8) | 119.9 (521.2) | 99.4 (41.4) | 7.1 (3.1) | 41.2 (3.2) | 345.1 (90.3) | 60.2 (17.9) |
| **Gender** | | | | | | | | | | |
| Male | 1072 (49.2) | 457 (42.6) | 98.1 (412.2) | 130.2 (71.4) | 108.3 (567.2) | 104.1 (49.5) | 6.3 (2.9) | 42.0 (3.2) | 361.1 (83.5) | 69.4 (19.2) |
| Female | 1105 (50.8) | 464 (42.0) | 55.3 (298.7) | 89.2 (59.6) | 62.4 (291.2) | 80.3 (30.2) | 5.9 (2.6) | 41.6 (3.0) | 322.0 (88.1) | 70.0 (19.3) |
| **Race/Ethnicity** | | | | | | | | | | |
| Mexican American | 169 (7.8) | 58 (34.3) | 120.1 (435.4) | 111.7 (69.9) | 142.7 (542.6) | 85.6 (47.9) | 5.9 (2.9) | 41.8 (3.3) | 319.9 (83.5) | 75.8 (19.6) |
| Other Hispanics | 188 (8.6) | 69 (36.7) | 69.0 (208.9) | 108.3 (58.1) | 55.2 (148.1) | 83.7 (29.8) | 5.9 (2.5) | 41.6 (3.3) | 326.0 (82.9) | 72.8 (18.1) |
| Non-Hispanic White | 1151 (52.9) | 517 (44.9) | 51.3 (261.2) | 101.6 (62.0) | 64.2 (369.6) | 90.7 (35.3) | 6.4 (2.7) | 41.8 (2.9) | 337.3 (87.2) | 66.9 (17.8) |
| Non-Hispanic Black | 439 (20.2) | 195 (44.4) | 122.5 (541.4) | 140.3 (84.9) | 111.1 (483.4) | 104.9 (57.5) | 5.8 (3.1) | 41.3 (3.3) | 364.5 (90.2) | 71.6 (22.4) |
| Non-Hispanic Asian | 196 (9.0) | 64 (32.7) | 92.2 (399.7) | 86.4 (54.0) | 132.6 (794.8) | 82.7 (41.9) | 5.7 (2.1) | 42.7 (2.9) | 342.3 (82.4) | 75.1 (17.3) |
| Other races | 34 (1.6) | 18 (52.9) | 60.4 (207.1) | 103.4 (56.7) | 55.4 (184.2) | 102.7 (35.1) | 5.8 (2.5) | 41.6 (3.3) | 361.3 (108.8) | 62.2 (21.3) |
| **Education** | | | | | | | | | | |
| Below high school | 649 (29.8) | 318 (49.0) | 103.9 (440.4) | 114.1 (72.8) | 108.4 (437.3) | 95.4 (49.4) | 6.3 (3.3) | 41.4 (3.3) | 348.4 (90.9) | 68.8 (21.3) |
| High school | 504 (23.2) | 213 (42.3) | 70.3 (308.3) | 114.2 (72.8) | 71.2 (307.4) | 92.2 (40.3) | 6.0 (2.5) | 41.9 (3.0) | 346.7 (86.9) | 69.4 (18.3) |
| College or above | 1019 (46.8) | 386 (37.9) | 61.9 (324.5) | 104.0 (63.7) | 76.9 (512.5) | 89.8 (38.7) | 6.0 (2.5) | 42.0 (2.9) | 333.8 (86.1) | 70.5 (18.3) |
| Missing | 5 (0.2) | 4 (80.0) | 74.2 (70.3) | 122.8 (68.2) | 82.8 (90.9) | 99.5 (31.8) | 5.1 (2.3) | 41.2 (2.2) | 397.3 (93.2) | 67.1 (23.8) |
| **Income (PIR)** | | | | | | | | | | |
| Tertile 1 (0–1.87) | 928 (42.6) | 416 (44.8) | 88.8 (415.6) | 109.5 (70.3) | 90.8 (412.7) | 91.8 (41.5) | 6.1 (2.9) | 41.7 (3.0) | 343.2 (89.8) | 69.8 (20.1) |
| Tertile 2 (1.88–3.86) | 582 (26.7) | 251 (43.1) | 65.8 (264.0) | 112.8 (72.3) | 76.0 (362.0) | 93.7 (47.5) | 6.1 (2.7) | 41.6 (3.3) | 341.9 (85.1) | 68.4 (19.0) |
| Tertile 3 (> = 3.87) | 474 (21.8) | 176 (37.1) | 50.7 (258.0) | 105.4 (63.4) | 64.7 (507.6) | 91.1 (36.8) | 6.0 (2.4) | 42.1 (2.9) | 336.7 (87.4) | 70.6 (17.9) |
| Missing | 193 (8.9) | 78 (40.4) | 111.3 (503.1) | 108.7 (63.3) | 134.2 (656.5) | 90.4 (45.1) | 6.4 (3.0) | 42.2 (3.1) | 341.3 (90.0) | 71.3 (18.9) |

(*Continued*)

**Table 2.** (Continued)

| Characteristics | n (%) | CKD n (%) | Urinary albumin (mg/L) | Urinary creatinine (mg/dL) | Albumin creatinine ratio (mg/g) | Serum creatinine (μmol/L) | Blood urea nitrogen (mmol/L) | Plasma albumin (g/L) | Uric acid (umol/L) | eGFR (mL/min per 1.73 m²) |
|---|---|---|---|---|---|---|---|---|---|---|
| **Marital Status** | | | | | | | | | | |
| Married | 1173 (53.9) | 450 (38.4) | 77.4 (379.3) | 112.0 (69.6) | 91.4 (530.9) | 93.2 (46.0) | 6.1 (2.8) | 42.1 (2.9) | 343.3 (86.3) | 70.7 (18.4) |
| Separated | 44 (2.0) | 15 (34.1) | 134.2 (349.6) | 106.5 (67.0) | 146.6 (456.0) | 88.5 (33.4) | 5.8 (2.9) | 41.8 (3.4) | 330.2 (90.7) | 72.7 (19.1) |
| Divorced | 258 (11.9) | 112 (43.4) | 60.5 (200.8) | 114.6 (72.9) | 58.2 (221.0) | 90.9 (40.2) | 5.7 (2.6) | 41.8 (2.8) | 342.7 (93.0) | 70.8 (20.1) |
| Widowed | 562 (25.8) | 292 (52.0) | 73.2 (365.8) | 99.2 (64.4) | 79.1 (343.4) | 90.4 (37.4) | 6.4 (2.8) | 41.2 (3.4) | 337.4 (89.5) | 66.4 (20.0) |
| Never married | 100 (4.6) | 40 (40.0) | 120.7 (466.4) | 115.6 (67.7) | 113.6 (456.4) | 93.3 (42.4) | 5.9 (2.6) | 42.1 (3.0) | 337.3 (90.9) | 71.8 (21.8) |
| Living with partner | 39 (1.8) | 11 (28.2) | 18.3 (18.8) | 131.0 (68.2) | 15.5 (19.0) | 88.5 (23.5) | 5.9 (1.8) | 42.1 (3.1) | 345.6 (77.4) | 74.4 (15.9) |
| Missing | 1 (0.1) | 1 (100.0) | 14.6 (0.0) | 154.9 (0.0) | 9.5 (0.0) | 80.4 (0.0) | 6.8 (0.0) | 32.0 (0.0) | 410.4 (0.0) | 59.8 (0.0) |
| **Health condition** | | | | | | | | | | |
| Excellent | 158 (7.3) | 59 (37.3) | 29.2 (94.1) | 114.7 (70.8) | 36.5 (151.5) | 91.5 (39.2) | 6.3 (2.2) | 42.0 (3.0) | 337.8 (79.2) | 69.0 (16.4) |
| Very good | 515 (23.7) | 172 (33.4) | 34.6 (185.1) | 103.6 (63.3) | 31.7 (127.3) | 84.2 (23.3) | 5.8 (2.0) | 42.1 (2.9) | 327.8 (85.9) | 72.4 (16.7) |
| Good | 811 (37.3) | 336 (41.4) | 82.5 (392.1) | 108.4 (68.5) | 88.1 (443.8) | 92.5 (37.7) | 6.0 (2.6) | 42.0 (2.9) | 346.6 (85.1) | 69.4 (19.4) |
| Fair | 509 (23.4) | 264 (51.9) | 109.5 (446.2) | 115.1 (71.6) | 121.3 (511.6) | 96.4 (48.7) | 6.4 (3.2) | 41.4 (3.3) | 346.8 (94.3) | 68.5 (20.9) |
| Poor | 95 (4.4) | 49 (51.6) | 171.8 (585.5) | 112.7 (79.5) | 243.9 (1152.4) | 104.2 (95.1) | 6.2 (3.9) | 40.4 (4.3) | 342.7 (97.8) | 67.6 (23.4) |
| Missing | 89 (4.1) | 41 (46.1) | 55.2 (144.5) | 107.1 (68.7) | 74.0 (242.4) | 95.6 (41.5) | 6.3 (3.8) | 41.7 (2.6) | 343.4 (87.3) | 67.3 (20.3) |
| **Smoking status** | | | | | | | | | | |
| Never smoker | 1096 (50.3) | 445 (40.6) | 75.1 (393.7) | 102.1 (63.1) | 84.6 (487.0) | 88.9 (42.6) | 6.1 (2.6) | 41.8 (3.0) | 333.6 (87.3) | 69.9 (19.0) |
| Former smoker | 857 (39.4) | 379 (44.2) | 68.9 (292.8) | 116.5 (72.1) | 73.9 (353.1) | 94.4 (39.4) | 6.2 (2.7) | 41.9 (3.1) | 350.4 (90.1) | 69.0 (18.8) |
| Current smoker | 222 (10.2) | 96 (43.2) | 111.6 (414.9) | 117.2 (78.5) | 130.3 (572.4) | 98.4 (51.8) | 5.6 (3.5) | 41.5 (3.4) | 343.6 (80.4) | 71.6 (22.0) |
| Missing | 2 (0.1) | 1 (50.0) | 64.7 (74.0) | 206.0 (52.3) | 27.7 (28.9) | 83.1 (1.3) | 6.3 (1.3) | 41.0 (2.8) | 350.9 (50.5) | 75.3 (26.2) |
| **Drinking status** | | | | | | | | | | |
| Never drinker | 392 (18.0) | 177 (45.2) | 98.0 (522.1) | 103.0 (67.8) | 111.5 (530.0) | 91.2 (55.3) | 6.2 (3.0) | 41.6 (3.0) | 333.8 (90.0) | 68.5 (20.8) |
| Former drinker | 318 (14.6) | 153 (48.1) | 84.2 (306.1) | 105.3 (65.0) | 93.2 (340.4) | 89.3 (34.3) | 6.2 (2.9) | 41.5 (3.1) | 335.7 (90.8) | 68.1 (19.7) |
| Current drinker | 1356 (62.3) | 541 (39.9) | 70.4 (324.8) | 112.4 (70.1) | 77.1 (460.3) | 92.7 (40.3) | 6.0 (2.6) | 41.9 (3.1) | 344.7 (86.8) | 70.6 (18.6) |
| Missing | 111 (5.1) | 50 (45.1) | 50.1 (131.0) | 107.5 (65.7) | 64.0 (218.1) | 94.7 (38.9) | 6.2 (3.6) | 41.8 (2.8) | 342.3 (87.0) | 67.7 (20.1) |
| **Physical activity** | | | | | | | | | | |
| Yes | 868 (39.9) | 324 (37.3) | 63.6 (298.8) | 111.7 (70.6) | 78.3 (507.9) | 90.7 (41.5) | 5.9 (2.5) | 42.1 (2.9) | 341.9 (86.5) | 71.7 (18.7) |
| No | 1306 (60.0) | 596 (45.6) | 84.6 (394.9) | 107.9 (67.6) | 89.4 (406.4) | 92.9 (43.2) | 6.2 (2.9) | 41.6 (3.2) | 340.8 (89.2) | 68.4 (19.5) |

*(Continued)*

**Table 2.** (Continued)

| Characteristics | n (%) | CKD n (%) | Urinary albumin (mg/L) | Urinary creatinine (mg/dL) | Albumin creatinine ratio (mg/g) | Serum creatinine (μmol/L) | Blood urea nitrogen (mmol/L) | Plasma albumin (g/L) | Uric acid (umol/L) | eGFR (mL/min per 1.73 m²) |
|---|---|---|---|---|---|---|---|---|---|---|
| Missing | 3 (0.1) | 1 (33.3) | 198.6 (339.0) | 106.0 (53.9) | 120.3 (201.7) | 82.5 (11.9) | 5.1 (1.0) | 40.3 (2.9) | 343.0 (44.6) | 80.3 (9.7) |
| **Body mass index (kg/m²)** | | | | | | | | | | |
| Underweight (<18.5) | 36 (1.7) | 16 (44.4) | 79.0 (261.2) | 106.1 (67.9) | 81.2 (238.0) | 77.5 (20.2) | 5.2 (2.0) | 42.1 (4.1) | 270.8 (74.1) | 75.8 (16.3) |
| Normal (18.5–24.9) | 579 (26.6) | 228 (39.4) | 79.2 (388.4) | 97.7 (67.3) | 103.8 (601.0) | 89.3 (45.6) | 6.0 (2.7) | 42.3 (3.0) | 319.4 (84.0) | 71.8 (19.8) |
| Overweight (25.0–29.9) | 776 (35.7) | 316 (40.7) | 44.9 (171.6) | 110.0 (67.0) | 46.3 (190.6) | 91.7 (30.8) | 6.0 (2.3) | 42.2 (2.8) | 338.1 (84.1) | 69.3 (18.1) |
| Obese (> = 30) | 746 (34.3) | 335 (44.9) | 104.7 (470.7) | 117.6 (70.4) | 107.0 (507.8) | 94.8 (50.5) | 6.3 (3.1) | 41.1 (3.2) | 363.7 (89.6) | 68.4 (19.7) |
| Missing | 40 (1.8) | 26 (65.0) | 116.6 (253.3) | 116.7 (73.2) | 159.6 (404.3) | 100.0 (41.6) | 7.4 (4.1) | 40.2 (3.7) | 363.6 (84.4) | 65.3 (23.8) |
| **Hypertension** | | | | | | | | | | |
| Yes | 746 (34.3) | 371 (49.7) | 131.6 (526.2) | 102.2 (67.1) | 159 (696.9) | 94.6 (46.8) | 6.2 (2.9) | 41.8 (3.1) | 341 (86.7) | 68.4 (20.4) |
| No | 1431 (65.7) | 550 (38.4) | 47.6 (224.0) | 113.2 (69.4) | 46.5 (223.4) | 90.7 (40.0) | 6.1 (2.7) | 41.8 (3.1) | 341.4 (88.8) | 70.4 (18.6) |
| **Diabetes** | | | | | | | | | | |
| Yes | 526 (24.2) | 302 (57.4) | 177.7 (612.9) | 112.6 (69.2) | 210.1 (810.0) | 104.3 (62.3) | 6.9 (3.4) | 41.3 (3.2) | 357.8 (93.4) | 65.3 (21.8) |
| No | 1650 (75.8) | 618 (37.5) | 44.1 (216.2) | 108.4 (68.7) | 45.2 (225.8) | 88.1 (33.0) | 5.9 (2.5) | 42 (3.0) | 336 (85.6) | 71.2 (18.1) |
| Missing | 1 (0.0) | 1 (100.0) | 3.2 (0.0) | 74 (0.0) | 4.3 (0.0) | 99.9 (0.0) | 7.1 (0.0) | 37 (0.0) | 404.5 (0.0) | 47 (0.0) |

Abbreviations: SD = standard deviation, CKD = chronic kidney diseases, eGFR = estimated glomerular filtration rate, PIR = ratio of family income to poverty.

There was a large difference in the urinary albumin between the two samples. The mean of urinary albumin level of CLHLS participants was 25.0 mg/L (SD: 75.4), much lower than that of NHANES participants (76.4 mg/L, SD: 359.6). Among Chinese participants, higher urinary albumin level usually appeared along with factors like age over 80 (29.5 mg/L), female (25.8 mg/L), high household income (29.8 mg/L), widowed (28.6 mg/L), good health condition (25.7 mg/L), never smoker (26.6 mg/L), and former drinker (34.7 mg/L) (Table 1). However, among NHANES participants, people had higher urinary albumin were more likely to be aged 75–79 (109.3 mg/L), male (98.1 mg/L), with low household income (88.8 mg/L), separated (134.2 mg/L), with poor health condition (171.8 mg/L), current smoker (111.6 mg/L), and never drinker (98.0 mg/L) (Table 2). Besides, some characteristics were found related to higher urinary albumin level in both samples, including lower education, obese, insufficient physical activity, hypertension, and diabetes (Tables 1 and 2).

In addition to urinary albumin, the mean level of ACR in Chinese participants (41.7 mg/g, SD: 231.5) was also significantly lower than that in the US (85.0 mg/g, SD: 449.3). In both countries, the subgroups with the lowest ACR were those were aged 70–74, with formal education, drinking currently, overweight, or without hypertension. However, we did not find a similar pattern in the two samples in terms of the relationship between ACR and other demographic or lifestyle characteristics.

Moreover, Chinese participants (292.9 μmol/L, SD: 90.4) also showed a lower mean level of uric acid than the US (341.3 μmol/L, SD: 88.0). Besides, male presented a higher mean level than female in both Chinese (male: 314.8 μmol/L, female: 274.0 μmol/L) and the US participants (male: 361.1 μmol/L, female: 322.0 μmol/L). Also, people who were younger, never smoker, never drinker, not taking physical activity, underweight, without diabetes tended to have lower uric acid in both CLHLS and NHANES group.

### Factors associated with CKD

There were different factors associated with CKD between CLHLS and NHANES samples. In China, CKD was found more common in participants who were older, female, with higher household income, self-rated bad health, or had physical activities. However, in the US, participants with older age, education below high school, self-rated bad health, belonging to white or other races, were not married, or currently smoking were more likely to have CKD. The effect size of risk factors for CKD was also different between CLHLS and NHANES. Compared to those aged 65–69, the ORs (95% CI) of 70–74, 75–80, and 80+ years old for CKD in CLHLS was 0.918 (0.568, 1.484), 2.080 (1.322, 3.300), and 4.187 (2.839, 6.283), lower than those in NHANES, as 1.492 (1.154, 1.928), 2.45 (1.839, 3.263), and 4.557 (3.441, 6.034) (Tables 3 and 4). In the Chinese sample, women were more closely related to CKD (OR: 1.625, 95% CI: 1.266, 2.088) compared to men, while the same pattern did not present in the US participants. US participants who were white had greater OR of CKD (1.555) than the Mexican American, and those with college or above educational level had a lower prevalence of CKD, compared to participants with educational level below high school. In CLHLS, the presence of CKD was found associated with higher household income, which was opposite to the situation in the NHANES participants. In both countries, ORs of CKD increased with worse health condition, but insignificantly in most cases. Only participants in the US who rated their health condition as "Fair" or "Poor" had a significant association with CKD (OR: 1.533, 95% CI: 1.024, 2.295). Among NHANES participants, current smokers also showed about 40% greater odds of having CKD. Besides, in the Chinese sample, those who had no physical activities had lower odds of CKD (0.708, 95% CI: 0.534, 0.938) than the others.

In the age stratified analysis, the association between most of the above risk factors and CKD only persisted in participants aged 80 or older in CLHLS. Former drinker was associated with higher odds of CKD only in Chinese participants aged younger than 80 (Table 3). The CKD risk factors were also different among different age groups in NHANES except hypertension and diabetes. Education and health condition were still significantly associated with CKD only in participants aged 80 or older in NHANES. Current smoker had a higher odds of CKD only in those aged 65–69 in NHANES (Table 4).

Most risk factors associated with CKD were also associated with abnormal eGFR and ACR when using abnormal eGFR or ACR as the dependent variable separately. Of note, obesity was risk factor for abnormal eGFR but not for CKD or abnormal ACR in NHANES. Older age was associated with CKD and abnormal eGFR but not with abnormal ACR in CLHLS (S5 and S6 Tables).

### The mortality risk of CKD biomarkers

The CLHLS participants were followed up from 2012 to 2018, with a total of 6307 person-years and 817 deaths. The participants in NHANES were followed up in 2015, with 6059 person-year and 201 deaths. In both CLHLS and NHANES, the level of BUN was positively correlated with all-cause mortality risk in the elderly [HR (95%CI): 1.041 (1.008, 1.075) in CLHLS, 1.106 (1.070, 1.143) in NHANES], plasma albumin was negatively associated with all-cause

**Table 3. Odds ratio (95% CI) of factors associated with CKD in Chinese participants (CLHLS 2012).**

| Factors | Total population | | | Age<80 | | | Age≥80 | | |
|---|---|---|---|---|---|---|---|---|---|
| | mean(sd) / n (%) | OR (95% CI) * | P value | mean(sd) / n (%) | OR (95% CI) * | P value | mean(sd) / n (%) | OR (95% CI) * | P value |
| Total | 2019 (100) | | | | | | | | |
| Age (mean ± SD) | 85.7 (12.2) | \ | \ | 71.8 (4.29) | **1.09 (1.04, 1.14)** | **<0.001** | 93.0 (7.88) | **1.04 (1.03, 1.06)** | **<0.001** |
| **Age group** | | | | | | | | | |
| 65–69 | 240 (11.9) | Ref | \ | 240 (34.4) | \ | \ | 0 (0) | \ | \ |
| 70–74 | 240 (11.9) | 0.92 (0.57, 1.48) | 0.727 | 240 (34.4) | \ | \ | 0 (0) | \ | \ |
| 75–79 | 217 (10.7) | **2.08 (1.32, 3.30)** | **0.002** | 217 (31.1) | \ | \ | 0 (0) | \ | \ |
| 80+ | 1322 (65.5) | **4.19 (2.84, 6.28)** | **<0.001** | 0 (0) | \ | \ | 1322 (100) | \ | \ |
| **Gender** | | | | | | | | | |
| Male | 933 (46.2) | Ref | \ | 457 (65.6) | Ref | \ | 476 (36.0) | Ref | \ |
| Female | 1086 (53.8) | **1.63 (1.27, 2.09)** | **<0.001** | 240 (34.4) | 1.53 (0.95, 2.48) | 0.084 | 846 (64.0) | **1.61 (1.19, 2.19)** | **0.002** |
| **Race** | | | | | | | | | |
| Han Chinese | 1817 (90.0) | Ref | \ | 622 (89.2) | Ref | \ | 1195 (90.4) | Ref | \ |
| Ethnic minorities | 152 (7.5) | 1.21 (0.84, 1.76) | 0.304 | 62 (8.9) | 1.07 (0.54, 2.02) | 0.849 | 90 (6.8) | 1.25 (0.79, 2.01) | 0.342 |
| Missing | 50 (2.5) | 1.31 (0.71, 2.46) | 0.389 | 13 (1.9) | 2.57 (0.70, 8.69) | 0.134 | 37 (2.8) | 1.17 (0.58, 2.43) | 0.663 |
| **Education** | | | | | | | | | |
| No formal education | 1238 (61.3) | Ref | \ | 239 (34.3) | Ref | \ | 999 (75.6) | Ref | \ |
| Formal education | 764 (37.8) | 1.06 (0.82, 1.38) | 0.654 | 456 (65.4) | 1.22 (0.78, 1.92) | 0.392 | 308 (23.3) | 1.16 (0.83, 1.61) | 0.388 |
| Missing | 17 (0.8) | 1.61 (0.55, 5.19) | 0.396 | 2 (0.3) | 4.02 (0.14, 111.83) | 0.352 | 15 (1.1) | 1.25 (0.40, 4.40) | 0.713 |
| **Household income (RMB)** | | | | | | | | | |
| Tertile 1 (<6,000) | 637 (31.6) | Ref | \ | 233 (33.4) | Ref | \ | 404 (30.6) | Ref | \ |
| Tertile 2 (6,000–19,000) | 661 (32.7) | **1.36 (1.07, 1.74)** | **0.012** | 236 (33.9) | 0.99 (0.62, 1.59) | 0.966 | 425 (32.1) | **1.50 (1.12, 2.00)** | **0.006** |
| Tertile 3 (20,000-over 100,000) | 572 (28.3) | **1.89 (1.46, 2.44)** | **<0.001** | 202 (29.0) | 1.00 (0.61, 1.66) | 0.986 | 370 (28.0) | **2.32 (1.70, 3.17)** | **<0.001** |
| Missing | 149 (7.4) | **1.64 (1.06, 2.56)** | **0.029** | 26 (3.7) | 1.10 (0.36, 2.98) | 0.856 | 123 (9.3) | **1.85 (1.13, 3.07)** | **0.016** |
| **Marital Status** | | | | | | | | | |
| Married | 774 (38.3) | Ref | \ | 499 (71.6) | Ref | \ | 275 (20.8) | Ref | \ |
| Not married | 1196 (59.2) | 1.25 (0.98, 1.60) | 0.075 | 189 (27.1) | 0.77 (0.49, 1.21) | 0.262 | 1007 (76.2) | 1.22 (0.88, 1.69) | 0.226 |
| Missing | 49 (2.4) | 0.75 (0.22, 2.39) | 0.636 | 9 (1.3) | 0.63 (0.03, 4.77) | 0.708 | 40 (3.0) | 0.51 (0.11, 2.24) | 0.378 |
| **Health condition** | | | | | | | | | |
| Very good | 103 (5.1) | Ref | \ | 48 (6.9) | Ref | \ | 55 (4.2) | Ref | \ |
| Good | 750 (37.1) | 0.88 (0.55, 1.42) | 0.592 | 312 (44.8) | 1.07 (0.49, 2.55) | 0.876 | 438 (33.1) | 0.95 (0.51, 1.74) | 0.866 |
| Fair | 775 (38.4) | 1.25 (0.78, 2.01) | 0.365 | 260 (37.3) | 1.51 (0.68, 3.65) | 0.333 | 515 (39.0) | 1.35 (0.73, 2.47) | 0.331 |
| Bad/Very bad | 213 (10.6) | 1.33 (0.78, 2.28) | 0.294 | 69 (9.9) | 2.03 (0.80, 5.46) | 0.144 | 144 (10.9) | 1.40 (0.71, 2.75) | 0.333 |

(*Continued*)

**Table 3.** (Continued)

| Factors | Total population | | | Age<80 | | | Age≥80 | | |
|---|---|---|---|---|---|---|---|---|---|
| | mean(sd) / n (%) | OR (95% CI) * | P value | mean(sd) / n (%) | OR (95% CI) * | P value | mean(sd) / n (%) | OR (95% CI) * | P value |
| Missing | 178 (8.8) | 0.90 (0.51, 1.60) | 0.716 | 8 (1.1) | 0.64 (0.02, 7.00) | 0.743 | 170 (12.9) | 0.82 (0.41, 1.60) | 0.557 |
| **Smoking status** | | | | | | | | | |
| Never smoker | 1465 (72.6) | Ref | \ | 432 (62.0) | Ref | \ | 1033 (78.1) | Ref | \ |
| Former smoker | 164 (8.1) | 0.75 (0.51, 1.12) | 0.160 | 58 (8.3) | 0.92 (0.43, 1.87) | 0.822 | 106 (8.0) | 0.68 (0.42, 1.09) | 0.106 |
| Current smoker | 334 (16.5) | 1.00 (0.73, 1.37) | 0.990 | 197 (28.3) | 0.92 (0.55, 1.53) | 0.741 | 137 (10.4) | 1.08 (0.71, 1.66) | 0.709 |
| Missing | 56 (2.8) | 1.02 (0.36, 2.84) | 0.970 | 10 (1.4) | 0.48 (0.02, 3.87) | 0.564 | 46 (3.5) | 1.28 (0.36, 4.72) | 0.700 |
| **Drinking status** | | | | | | | | | |
| Never drinker | 1528 (75.7) | Ref | \ | 495 (71.0) | Ref | \ | 1033 (78.1) | Ref | \ |
| Former drinker | 120 (5.9) | 1.25 (0.81, 1.93) | 0.321 | 44 (6.3) | **2.29 (1.09, 4.71)** | **0.026** | 76 (5.7) | 1.03 (0.60, 1.76) | 0.927 |
| Current drinker | 315 (15.6) | 0.87 (0.64, 1.18) | 0.368 | 146 (20.9) | 0.94 (0.53, 1.64) | 0.832 | 169 (12.8) | 0.83 (0.57, 1.21) | 0.333 |
| Missing | 56 (2.8) | 1.61 (0.56, 4.59) | 0.367 | 12 (1.7) | 1.00 (0.11, 5.63) | 0.997 | 44 (3.3) | 2.36 (0.57, 11.75) | 0.258 |
| **Physical activity** | | | | | | | | | |
| Yes | 311 (15.4) | Ref | \ | 137 (19.7) | Ref | \ | 174 (13.2) | Ref | \ |
| No | 1598 (79.1) | **0.71 (0.53, 0.94)** | **0.016** | 533 (76.5) | **0.58 (0.36, 0.94)** | **0.024** | 1065 (80.6) | **0.68 (0.47, 0.98)** | **0.038** |
| Missing | 110 (5.4) | 0.68 (0.38, 1.20) | 0.183 | 27 (3.9) | 0.69 (0.20, 2.03) | 0.527 | 83 (6.3) | 0.69 (0.34, 1.37) | 0.286 |
| **Body mass index (kg/m²)** | | | | | | | | | |
| Underweight (<18.5) | 477 (23.6) | Ref | \ | 70 (10.0) | Ref | \ | 407 (30.8) | Ref | \ |
| Normal (18.5–24.9) | 1153 (57.1) | **0.74 (0.58, 0.94)** | **0.013** | 465 (66.7) | 0.66 (0.36, 1.24) | 0.186 | 688 (52.0) | 0.83 (0.63, 1.08) | 0.161 |
| Overweight (25.0–29.9) | 229 (11.3) | **0.60 (0.41, 0.87)** | **0.008** | 132 (18.9) | 0.64 (0.31, 1.36) | 0.243 | 97 (7.3) | 0.63 (0.39, 1.02) | 0.063 |
| Obese (> = 30) | 58 (2.9) | 0.67 (0.36, 1.23) | 0.199 | 24 (3.4) | 0.31 (0.06, 1.12) | 0.098 | 34 (2.6) | 0.90 (0.42, 1.98) | 0.797 |
| Missing | 102 (5.1) | 0.88 (0.55, 1.41) | 0.586 | 6 (0.9) | 3.31 (0.51, 22.08) | 0.197 | 96 (7.3) | 0.75 (0.46, 1.23) | 0.247 |
| **Hypertension** | | | | | | | | | |
| yes | 1142 (56.6) | Ref | \ | 342 (49.1) | Ref | \ | 800 (60.5) | Ref | \ |
| no | 857 (42.4) | **0.71 (0.58, 0.87)** | **0.001** | 350 (50.2) | 0.77 (0.52, 1.14) | 0.192 | 507 (38.4) | **0.67 (0.53, 0.86)** | **0.001** |
| missing | 20 (1.0) | 1.41 (0.54, 3.93) | 0.492 | 5 (0.7) | 0.25 (0.01, 2.66) | 0.318 | 15 (1.1) | 1.90 (0.60, 7.33) | 0.303 |
| **Diabetes** | | | | | | | | | |
| yes | 48 (2.4) | Ref | \ | 29 (4.2) | Ref | \ | 19 (1.4) | Ref | \ |
| no | 1940 (96.1) | 0.62 (0.33, 1.19) | 0.150 | 665 (95.4) | 0.66 (0.28, 1.64) | 0.348 | 1275 (96.4) | 0.36 (0.12, 0.99) | 0.056 |
| missing | 31 (1.5) | 0.57 (0.21, 1.57) | 0.274 | 3 (0.4) | 1.24 (0.05, 19.36) | 0.877 | 28 (2.1) | 0.29 (0.08, 1.07) | 0.065 |

Abbreviations: OR = odds ratio, CI = confidence interval, CKD = chronic kidney diseases.

* The multi-variate analysis contained all the variables listed above in the logistic regression models.

**Table 4. Odds ratio (95% CI) of factors associated with CKD in US participants (NHANES 2011–2014).**

| Factors | Total | | | Age: 65–69 | | | Age: 70–74 | | | Age: 75–79 | | | Age: 80+ | | |
|---|---|---|---|---|---|---|---|---|---|---|---|---|---|---|---|
| | mean (sd) / n (%) | OR (95% CI) [*] | P value | mean (sd) / n (%) | OR (95% CI) [*] | P value | mean (sd) / n (%) | OR (95% CI) [*] | P value | mean (sd) / n (%) | OR (95% CI) [a] | P value | mean (sd) / n (%) | OR (95% CI) [a] | P value |
| **Total** | 2177 (100) | \ | \ | 682 (100) | \ | \ | 567 (100) | \ | \ | 361 (100) | \ | \ | 567 (100) | \ | \ |
| **Age** | 73.2 (5.4) | \ | \ | 66.9 (1.4) | 1.12 (0.99, 1.28) | 0.082 | 71.9 (1.4) | **1.15 (1.01, 1.31)** | **0.034** | 76.8 (1.4) | 1.04 (0.89, 1.23) | 0.612 | 80.0(0)[b] | NA | NA |
| **Age group** | | | | | | | | | | | | | | | |
| 65–69 | 682 (31.3) | Ref | \ | 682 (100) | \ | \ | 0 | \ | \ | 0 | \ | \ | 0 | \ | \ |
| 70–74 | 567 (26.1) | **1.49 (1.15, 1.93)** | **0.002** | 0 | \ | \ | 567 (100) | \ | \ | 0 | \ | \ | 0 | \ | \ |
| 75–79 | 361 (16.6) | **2.45 (1.84, 3.26)** | **< .001** | 0 | \ | \ | 0 | \ | \ | 361 (100) | \ | \ | 0 | | \ |
| 80+ | 567 (26.1) | **4.56 (3.44, 6.03)** | **< .001** | 0 | \ | \ | 0 | \ | \ | 0 | | \ | 567 (100) | \ | \ |
| **Gender** | | | | | | | | | | | | | | | |
| Male | 1072 (49.2) | Ref | \ | 341 (50.0) | Ref | \ | 264 (46.6) | Ref | \ | 190 (52.6) | Ref | \ | 277 (48.9) | Ref | \ |
| Female | 1105 (50.8) | 0.94 (0.76, 1.16) | 0.543 | 341 (50.0) | 0.92 (0.61, 1.39) | 0.682 | 303 (53.4) | 0.91 (0.60, 1.39) | 0.658 | 171 (47.4) | 1.12 (0.66, 1.91) | 0.667 | 290 (51.2) | 0.95 (0.63, 1.44) | 0.813 |
| **Race/Ethnicity** | | | | | | | | | | | | | | | |
| Mexican American | 169 (7.8) | Ref | \ | 84 (12.3) | Ref | \ | 44 (7.8) | Ref | \ | 23 (6.4) | Ref | \ | 18 (3.2) | Ref | \ |
| Other Hispanics | 188 (8.6) | 1.07 (0.67, 1.72) | 0.770 | 86 (12.6) | 1.28 (0.60, 2.71) | 0.519 | 50 (8.8) | 1.56 (0.61, 4.02) | 0.355 | 25 (6.9) | 0.29 (0.08, 1.11) | 0.070 | 27 (4.8) | 0.72 (0.18, 2.96) | 0.653 |
| Non-Hispanic White | 1151 (52.9) | **1.56 (1.05, 2.31)** | **0.028** | 234 (34.3) | 1.43 (0.73, 2.80) | 0.302 | 304 (53.6) | 2.14 (0.95, 4.85) | 0.068 | 193 (53.5) | 0.59 (0.20, 1.72) | 0.331 | 420 (74.1) | 0.99 (0.31, 3.18) | 0.985 |
| Non-Hispanic Black | 439 (20.2) | 1.49 (0.99, 2.24) | 0.054 | 183 (26.8) | 1.57 (0.80, 3.06) | 0.187 | 103 (18.2) | **2.45 (1.04, 5.77)** | **0.041** | 87 (24.1) | 0.61 (0.20, 1.85) | 0.385 | 66 (11.6) | 0.67 (0.19, 2.32) | 0.529 |
| Non-Hispanic Asian | 196 (9.0) | 1.12 (0.68, 1.82) | 0.664 | 85 (12.5) | 1.44 (0.64, 3.26) | 0.378 | 55 (9.7) | 0.90 (0.31, 2.58) | 0.841 | 26 (7.2) | 0.70 (0.19, 2.59) | 0.592 | 30 (5.3) | 0.62 (0.15, 2.49) | 0.496 |
| Other races | 34 (1.6) | **2.28 (1.02, 5.09)** | **0.045** | 10 (1.5) | **5.78 (1.36, 24.51)** | **0.017** | 11 (1.9) | 2.93 (0.68, 12.59) | 0.148 | 7 (1.9) | 0.69 (0.10, 4.59) | 0.701 | 6 (1.1) | 0.75 (0.09, 6.50) | 0.792 |
| **Education** | | | | | | | | | | | | | | | |
| Below high school | 649 (29.8) | Ref | \ | 200 (29.3) | Ref | \ | 161 (28.4) | Ref | \ | 116 (32.1) | Ref | \ | 172 (30.3) | Ref | \ |
| High school | 504 (23.2) | 0.80 (0.61, 1.04) | 0.097 | 147 (21.6) | 1.10 (0.65, 1.86) | 0.732 | 136 (24) | 0.77 (0.45, 1.32) | 0.340 | 83 (23) | 1.11 (0.57, 2.15) | 0.760 | 138 (24.3) | **0.51 (0.30, 0.87)** | **0.012** |
| College or above | 1019 (46.8) | **0.78 (0.60, 0.996)** | **0.047** | 334 (49.0) | 0.90 (0.55, 1.49) | 0.689 | 269 (47.4) | 0.88 (0.52, 1.48) | 0.624 | 162 (44.9) | 1.04 (0.56, 1.94) | 0.908 | 254 (44.8) | **0.54 (0.33, 0.89)** | **0.017** |

(*Continued*)

**Table 4.** (Continued)

| Factors | Total | | | Age: 65–69 | | | Age: 70–74 | | | Age: 75–79 | | | Age: 80+ | | |
|---|---|---|---|---|---|---|---|---|---|---|---|---|---|---|---|
| | mean (sd) / n (%) | OR (95% CI) * | P value | mean (sd) / n (%) | OR (95% CI) * | P value | mean (sd) / n (%) | OR (95% CI) * | P value | mean (sd) / n (%) | OR (95% CI) ᵃ | P value | mean (sd) / n (%) | OR (95% CI) ᵃ | P value |
| Missing | 5 (0.2) | 2.37 (0.26, 21.84) | 0.448 | 1 (0.2) | NA | 0.996 | 1 (0.2) | NA | 0.980 | 0 | \ | \ | 3 (0.5) | 1.40 (0.11, 17.16) | 0.793 |
| **Income (PIR)** | | | | | | | | | | | | | | | |
| Tertile 1 (0–1.87) | 928 (42.6) | Ref | \ | 294 (43.1) | Ref | \ | 230 (40.6) | Ref | \ | 158 (43.8) | Ref | \ | 246 (43.4) | Ref | \ |
| Tertile 2 (1.88–3.86) | 582 (26.7) | 1.14 (0.89, 1.44) | 0.298 | 171 (25.1) | 1.12 (0.69, 1.81) | 0.646 | 148 (26.1) | 1.16 (0.70, 1.93) | 0.553 | 104 (28.8) | 0.66 (0.38, 1.17) | 0.154 | 159 (28) | **1.60 (1.00, 2.57)** | **0.050** |
| Tertile (> = 3.87) | 474 (21.8) | 1.13 (0.86, 1.50) | 0.382 | 163 (24.0) | 1.11 (0.63, 1.97) | 0.714 | 134 (23.6) | 1.36 (0.77, 2.38) | 0.290 | 68 (18.8) | 0.86 (0.43, 1.72) | 0.674 | 109 (19.2) | 1.15 (0.67, 1.98) | 0.606 |
| Missing | 193 (8.9) | 0.90 (0.63, 1.27) | 0.530 | 54 (7.9) | 0.80 (0.38, 1.70) | 0.559 | 55 (9.7) | 1.11 (0.56, 2.19) | 0.772 | 31 (8.6) | 0.75 (0.33, 1.73) | 0.498 | 53 (9.4) | 0.83 (0.43, 1.61) | 0.587 |
| **Marital Status** | | | | | | | | | | | | | | | |
| Married | 1173 (53.9) | Ref | \ | 383 (56.2) | Ref | \ | 345 (60.9) | Ref | \ | 197 (54.6) | Ref | \ | 248 (43.7) | Ref | \ |
| Not married | 1003 (46.1) | 1.21 (0.99, 1.49) | 0.061 | 299 (43.8) | 0.92 (0.61, 1.39) | 0.679 | 222 (39.2) | **1.65 (1.09, 2.49)** | **0.017** | 164 (45.4) | 0.92 (0.55, 1.54) | 0.754 | 318 (56.1) | 1.14 (0.76, 1.69) | 0.534 |
| Missing | 1 (0.1) | NA | 0.978 | 0 | \ | \ | 0 | \ | \ | 0 | | \ | 1 (0.2) | NA | 0.986 |
| **Health condition** | | | | | | | | | | | | | | | |
| Excellent | 158 (7.3) | Ref | \ | 45 (6.6) | Ref | \ | 46 (8.1) | Ref | \ | 22 (6.1) | Ref | \ | 45 (7.9) | Ref | \ |
| Very good | 515 (23.7) | 0.78 (0.52, 1.15) | 0.209 | 154 (22.6) | 0.52 (0.23, 1.17) | 0.114 | 141 (24.9) | 0.57 (0.27, 1.21) | 0.141 | 82 (22.7) | 0.57 (0.20, 1.61) | 0.290 | 138 (24.3) | 1.28 (0.63, 2.59) | 0.495 |
| Good | 811 (37.3) | 1.05 (0.72, 1.54) | 0.810 | 261 (38.3) | 0.57 (0.26, 1.24) | 0.154 | 207 (36.5) | 0.95 (0.47, 1.93) | 0.888 | 136 (37.7) | 0.66 (0.25, 1.80) | 0.422 | 207 (36.5) | **2.14 (1.07, 4.29)** | **0.032** |
| Fair/Poor | 604 (27.7) | **1.53 (1.02, 2.30)** | **0.038** | 204 (29.9) | 0.89 (0.39, 2.01) | 0.779 | 154 (27.2) | 1.46 (0.69, 3.09) | 0.317 | 98 (27.2) | 1.18 (0.40, 3.45) | 0.760 | 148 (26.1) | **3.00 (1.40, 6.43)** | **0.005** |
| Missing | 89 (4.1) | 2.26 (0.78, 6.53) | 0.132 | 18 (2.6) | 1.47 (0.10, 21.63) | 0.780 | 19 (3.4) | NA | 0.980 | 23 (6.4) | 3.48 (0.15, 81.36) | 0.438 | 29 (5.1) | 4.07 (0.91, 18.20) | 0.067 |
| **Smoking status** | | | | | | | | | | | | | | | |
| Never smoker | 1096 (50.3) | Ref | \ | 317 (46.5) | Ref | \ | 298 (52.6) | Ref | \ | 163 (45.2) | Ref | \ | 318 (56.1) | Ref | \ |
| Former smoker | 857 (39.4) | 1.13 (0.91, 1.40) | 0.279 | 251 (36.8) | **1.72 (1.10, 2.68)** | **0.018** | 212 (37.4) | 0.71 (0.46, 1.10) | 0.123 | 165 (45.7) | 1.31 (0.77, 2.23) | 0.324 | 229 (40.4) | 1.09 (0.72, 1.65) | 0.699 |
| Current smoker | 222 (10.2) | **1.48 (1.05, 2.08)** | **0.025** | 114 (16.7) | **1.85 (1.05, 3.26)** | **0.034** | 56 (9.9) | 0.91 (0.45, 1.82) | 0.785 | 33 (9.1) | 2.20 (0.90, 5.41) | 0.085 | 19 (3.4) | 2.26 (0.65, 7.81) | 0.198 |
| Missing | 2 (0.1) | 0.67 (0.03, 16.00) | 0.802 | 0 | \ | \ | 1 (0.2) | NA | 0.980 | 0 | \ | \ | 1 (0.2) | NA | 0.988 |
| **Drinking status** | | | | | | | | | | | | | | | |

(*Continued*)

**Table 4.** (Continued)

| Factors | Total | | | Age: 65–69 | | | Age: 70–74 | | | Age: 75–79 | | | Age: 80+ | | |
|---|---|---|---|---|---|---|---|---|---|---|---|---|---|---|---|
| | mean (sd) / n (%) | OR (95% CI) * | P value | mean (sd) / n (%) | OR (95% CI) * | P value | mean (sd) / n (%) | OR (95% CI) * | P value | mean (sd) / n (%) | OR (95% CI) a | P value | mean (sd) / n (%) | OR (95% CI) a | P value |
| Never drinker | 392 (18.0) | Ref | \ | 100 (14.7) | Ref | \ | 97 (17.1) | Ref | \ | 70 (19.4) | Ref | \ | 125 (22.1) | Ref | \ |
| Former drinker | 318 (14.6) | 1.16 (0.83, 1.61) | 0.383 | 87 (12.8) | 1.44 (0.71, 2.92) | 0.311 | 90 (15.9) | 1.13 (0.59, 2.17) | 0.720 | 48 (13.3) | 1.02 (0.45, 2.30) | 0.967 | 93 (16.4) | 1.15 (0.62, 2.13) | 0.656 |
| Current drinker | 1356 (62.3) | 0.89 (0.67, 1.18) | 0.420 | 471 (69.1) | 0.78 (0.42, 1.42) | 0.412 | 360 (63.5) | 1.12 (0.63, 2.01) | 0.695 | 218 (60.4) | 0.53 (0.27, 1.06) | 0.072 | 307 (54.1) | 1.05 (0.63, 1.76) | 0.854 |
| Missing | 111 (5.1) | 0.49 (0.20, 1.23) | 0.129 | 24 (3.5) | 0.44 (0.04, 4.69) | 0.499 | 20 (3.5) | NA | 0.980 | 25 (6.9) | 0.14 (0.01, 2.65) | 0.188 | 42 (7.4) | 0.68 (0.21, 2.19) | 0.513 |
| **Physical activity** | | | | | | | | | | | | | | | |
| Yes | 868 (39.9) | Ref | \ | 334 (49.0) | Ref | \ | 244 (43.0) | Ref | \ | 140 (38.8) | Ref | \ | 150 (26.5) | Ref | \ |
| No | 1306 (60.0) | 1.02 (0.84, 1.25) | 0.810 | 347 (50.9) | 1.10 (0.76, 1.59) | 0.626 | 322 (56.8) | 0.96 (0.65, 1.41) | 0.822 | 220 (60.9) | 1.22 (0.76, 1.96) | 0.412 | 417 (73.5) | 0.89 (0.58, 1.35) | 0.571 |
| Missing | 3 (0.1) | 0.96 (0.08, 11.34) | 0.975 | 1 (0.2) | NA | 0.995 | 1 (0.2) | NA | 0.978 | 1 (0.3) | NA | 0.991 | 0 | \ | \ |
| **Body mass index (kg/m²)** | | | | | | | | | | | | | | | |
| Underweight (<18.5) | 36 (1.7) | Ref | \ | 8 (1.2) | Ref | \ | 9 (1.6) | Ref | \ | 4 (1.1) | Ref | \ | 15 (2.7) | Ref | \ |
| Normal (18.5–24.9) | 579 (26.6) | 1.02 (0.49, 2.14) | 0.958 | 178 (26.1) | 1.83 (0.20, 17.03) | 0.596 | 129 (22.8) | 0.65 (0.15, 2.76) | 0.557 | 92 (25.5) | 4.10 (0.34, 49.48) | 0.267 | 180 (31.8) | 0.75 (0.23, 2.51) | 0.643 |
| Overweight (25.0–29.9) | 776 (35.7) | 1.15 (0.55, 2.41) | 0.712 | 225 (33.0) | 1.69 (0.18, 15.90) | 0.645 | 208 (36.7) | 0.65 (0.16, 2.76) | 0.563 | 127 (35.2) | 6.98 (0.57, 85.61) | 0.129 | 216 (38.1) | 1.05 (0.31, 3.51) | 0.940 |
| Obese (> = 30) | 746 (34.3) | 1.33 (0.63, 2.81) | 0.451 | 265 (38.9) | 2.34 (0.25, 21.75) | 0.455 | 212 (37.4) | 0.90 (0.21, 3.83) | 0.889 | 134 (37.1) | 4.70 (0.38, 57.71) | 0.227 | 135 (23.8) | 1.24 (0.36, 4.29) | 0.732 |
| Missing | 40 (1.8) | 2.07 (0.75, 5.72) | 0.159 | 6 (0.9) | NA | 0.987 | 9 (1.6) | 2.87 (0.37, 22.42) | 0.314 | 4 (1.1) | 4.13 (0.15, 110.30) | 0.398 | 21 (3.7) | 2.65 (0.47, 15.12) | 0.272 |
| **Hypertension** | | | | | | | | | | | | | | | |
| Yes | 746 (34.3) | Ref | \ | 211 (30.9) | Ref | \ | 174 (30.7) | Ref | \ | 132 (36.6) | Ref | \ | 229 (40.4) | Ref | \ |
| No | 1431 (65.7) | **0.67 (0.55, 0.81)** | **<0.001** | 471 (69.1) | **0.46 (0.31, 0.68)** | **<0.001** | 393 (69.3) | 0.93 (0.62, 1.39) | 0.728 | 229 (63.4) | 0.83 (0.51, 1.35) | 0.453 | 338 (59.6) | **0.57 (0.39, 0.84)** | **0.004** |
| **Diabetes** | | | | | | | | | | | | | | | |
| Yes | 526 (24.2) | Ref | \ | 163 (23.9) | Ref | \ | 137 (24.2) | Ref | \ | 108 (29.9) | Ref | \ | 118 (20.8) | Ref | \ |
| No | 1650 (75.8) | **0.46 (0.37, 0.57)** | **<0.001** | 519 (76.1) | **0.33 (0.22, 0.51)** | **<0.001** | 430 (75.8) | **0.47 (0.30, 0.74)** | **0.001** | 252 (69.8) | **0.47 (0.27, 0.80)** | **0.005** | 449 (79.2) | **0.55 (0.33, 0.91)** | **0.020** |

(*Continued*)

**Table 4.** (Continued)

| Factors | Total | | | Age: 65–69 | | | Age: 70–74 | | | Age: 75–79 | | | Age: 80+ | | |
|---|---|---|---|---|---|---|---|---|---|---|---|---|---|---|---|
| | mean (sd) / n (%) | OR (95% CI) * | P value | mean (sd) / n (%) | OR (95% CI) * | P value | mean (sd) / n (%) | OR (95% CI) * | P value | mean (sd) / n (%) | OR (95% CI) a | P value | mean (sd) / n (%) | OR (95% CI) a | P value |
| Missing | 1 (0) | NA | 0.980 | 0 | \ | \ | 0 | \ | \ | 1 (0.3) | NA | 0.991 | 0 | \ | \ |

Abbreviations: OR = odds ratio, CI = confidence interval, CKD = chronic kidney diseases, PIR = ratio of family income to poverty.

* The multi-variate analysis contained all the variables listed above in the logistic regression models.

Abbreviations: OR = odds ratio, CI = confidence interval, CKD = chronic kidney diseases, PIR = ratio of family income to poverty.

a. The multi-variate analysis contained all the variables listed above in the logistic regression models.

b. In NHANES, the age of people over 80 years old was all coded as 80.

mortality [HR (95%CI): 0.971 (0.956, 0.986) in CLHLS, 0.893 (0.856, 0.933) in NHANES], and urinary creatinine was not significantly associated with mortality after adjusted for all covariates (Tables 5 and 6). The level of urinary albumin was related to higher mortality in CLHLS [HR (95%CI): 1.001 (1.000, 1.002)], whereas this association was statistically significant but clinically meaningless in NHANES [HR (95%CI): 1.000 (1.000, 1.000)]. In NHANES, increased uric acid was associated with greater odds of death [HR (95%CI): 1.003 (1.001, 1.004)], while the effect was not significant in CLHLS after adjusted for all covariates. Serum creatinine increased the mortality risk in both CLHLS and NHANES.

In both CLHLS and NHANES, the elderly with CKD had higher mortality risk than those without CKD [crude HRs (95% CI): 1.955 (1.703, 2.245) in CLHLS, 3.646 (2.679, 4.963) in NHANES] (Tables 5 and 6 and Fig 1). However, after adjusted for age and sex, the effect size diminished, and became insignificant in the Chinese group, while in the US group remained significant [CLHLS: 1.136 (0.983, 1.312, p>0.05), NHANES: 2.470 (1.796, 3.396, p<0.001)].

After stratified by the CKD stages, the population with low eGFR had higher odds of death than those with high eGFR in both CLHLS and NHANES (Tables 5 and 6 and Fig 2). After adjusted for all covariates, compared with the group with eGFR≥90 mL/min per 1.73 m$^2$, the HRs (95% CI) of the elderly whose eGFR under 30 remained significant in both CLHLS and NHANES [1.786 (1.047, 3.049) in CLHLS, 3.564 (1.712, 7.420) in NHANES], while eGFR of 30–45 mL/min per 1.73 m$^2$ did not increase mortality risk significantly in CLHLS [HRs (95% CI): 1.408 (0.884, 2.241)], but had doubled the mortality risk compared with those with eGFR≥90 in NHANES [HRs (95% CI): 2.249 (1.141, 4.430)]. Those with abnormal ACR (≥30) had higher mortality risk than those with normal ACR in both CLHLS and NHANES.

In the age-stratified analyses, biomarkers were associated with mortality risk only in participants aged 80 or older except for the abnormal ACR. In NHANES, most biomarkers were consistently associated with mortality risk in all age groups, but the effect were mostly less significant in the group aged 70–74.

## Discussion

In the current study, we found that the prevalence of CKD was 44.4% in the Chinese participants, and 42.3% in the NHANES sample. The level of ACR and eGFR in the US participants were unhealthier than the Chinese in most age groups. Older age, female, higher household income, and hypertension were found associated with CKD in the Chinese participants. In the US sample, gender was not associated with CKD, while high household income and low education level were associated with a higher prevalence of CKD. Furthermore, the association

**Table 5. Hazard ratio (95% CI) of biomarkers on mortality in Chinese participants (CLHLS 2012).**

| Model | Factor | CLHLS | | |
|---|---|---|---|---|
| | | Crude | Age-sex adjusted | All covariates adjusted † |
| **Total population** | | | | |
| Model A | Urinary albumin (mg/L) | **1.002 (1.001, 1.002)**\*\*\* | **1.001 (1.000, 1.002)**\* | **1.001 (1.000, 1.002)**\* |
| Model B | Urinary creatinine (mg/dL) | **0.997 (0.995, 0.998)**\*\*\* | 0.999 (0.998, 1.000) | 0.999 (0.998, 1.000) |
| Model C | Blood urea nitrogen (mmol/L) | **1.11 (1.07, 1.14)**\*\*\* | 1.03 (0.997, 1.06) | **1.04 (1.01, 1.08)**\* |
| Model D | Plasma albumin (g/L) | **0.92 (0.91, 0.94)**\*\*\* | **0.97 (0.96, 0.99)**\*\*\* | **0.97 (0.96, 0.99)**\*\*\* |
| Model E | Uric acid (umol/L) | 1.00 (0.999, 1.001) | 1.001 (1.000, 1.002) | 1.001 (1.000, 1.002) |
| Model F | Serum creatinine (μmol/L) | **1.004 (1.002, 1.006)**\*\*\* | 1.002 (1.000, 1.004) | **1.003 (1.000, 1.005)**\* |
| Model G | Albumin creatinine ratio (mg/g) | **1.0003 (1.0002, 1.0005)**\*\*\* | 1.00 (1.00, 1.00) | 1.00 (1.00, 1.00) |
| Model H | Categorical ACR | | | |
| | <30 | Ref | Ref | Ref |
| | ≥30 | **1.75 (1.50, 2.04)**\*\*\* | **1.24 (1.06, 1.45)**\*\* | **1.25 (1.07, 1.47)**\*\* |
| Model I | eGFR | **0.98 (0.97, 0.98)**\*\*\* | 0.997 (0.992, 1.001) | 0.996 (0.992, 1.001) |
| Model J | Categorical eGFR | | | |
| | <30 | **7.88 (4.80, 12.95)**\*\*\* | 1.45 (0.86, 2.46) | **1.79 (1.05, 3.05)**\* |
| | 30~ | **7.14 (4.65, 10.96)**\*\*\* | 1.36 (0.86, 2.15) | 1.41 (0.88, 2.24) |
| | 45~ | **5.15 (3.45, 7.70)**\*\*\* | 1.10 (0.71, 1.69) | 1.07 (0.69, 1.65) |
| | 60~ | **3.53 (2.39, 5.20)**\*\*\* | 1.10 (0.73, 1.66) | 1.16 (0.77, 1.75) |
| | 90~ | Ref | Ref | Ref |
| Model K | CKD | | | |
| | No | Ref | Ref | Ref |
| | Yes | **1.96 (1.70, 2.25)**\*\*\* | 1.14 (0.98, 1.31) | 1.09 (0.94, 1.27) |
| **Age<80** | | | | |
| Model A | Urinary albumin (mg/L) | 1.003 (1.000, 1.006) | 1.003 (0.999, 1.006) | 1.002 (0.998, 1.006) |
| Model B | Urinary creatinine (mg/dL) | 0.999 (0.996, 1.003) | 0.999 (0.995, 1.002) | 0.999 (0.995, 1.003) |
| Model C | Blood urea nitrogen (mmol/L) | 0.99 (0.87, 1.13) | 0.99 (0.87, 1.12) | 1.02 (0.88, 1.17) |
| Model D | Plasma albumin (g/L) | **0.94 (0.89, 0.99)**\* | 0.96 (0.91, 1.01) | 0.96 (0.90, 1.02) |
| Model E | Uric acid (umol/L) | 1.000 (0.998, 1.003) | 1.000 (0.997, 1.002) | 1.000 (0.997, 1.003) |
| Model F | Serum creatinine (μmol/L) | 1.01 (0.996, 1.02) | 1.001 (0.99, 1.01) | 0.998 (0.99, 1.01) |
| Model G | Albumin creatinine ratio (mg/g) | **1.004 (1.000, 1.007)**\* | **1.003 (1.000, 1.007)**\* | 1.004 (1.000, 1.008) |
| Model H | Categorical ACR | | | |
| | <30 | Ref | Ref | Ref |
| | ≥30 | **1.95 (1.09, 3.49)**\* | **1.94 (1.08, 3.48)**\* | **1.89 (1.01, 3.54)**\* |
| Model I | eGFR | 0.99 (0.98, 1.01) | 1.001 (0.99, 1.02) | 1.004 (0.99, 1.02) |
| Model J | Categorical eGFR | | | |
| | <30 | 2.33 (0.31, 17.57) | 1.53 (0.20, 11.71) | 1.06 (0.12, 9.48) |
| | 30~ | 1.00 (0.13, 7.55) | 0.55 (0.07, 4.20) | 0.43 (0.05, 3.71) |
| | 45~ | 1.34 (0.55, 3.25) | 0.80 (0.32, 2.02) | 0.77 (0.29, 2.05) |
| | 60~ | 1.37 (0.78, 2.40) | 0.92 (0.51, 1.68) | 0.94 (0.50, 1.74) |
| | 90~ | Ref | Ref | Ref |
| Model K | CKD | | | |
| | No | Ref | Ref | Ref |
| | Yes | 1.53 (0.92, 2.52) | 1.35 (0.81, 2.25) | 1.25 (0.72, 2.16) |
| **Age≥80** | | | | |
| Model A | Urinary albumin (mg/L) | **1.001 (1.000, 1.002)**\* | 1.001 (1.000, 1.001) | 1.001 (1.000, 1.002) |
| Model B | Urinary creatinine (mg/dL) | **0.998 (0.997, 0.999)**\*\* | 0.999 (0.998, 1.000) | 0.999 (0.998, 1.000) |
| Model C | Blood urea nitrogen (mmol/L) | **1.06 (1.03, 1.10)**\*\*\* | 1.03 (0.998, 1.06) | **1.04 (1.01, 1.08)**\* |

*(Continued)*

**Table 5.** (Continued)

| Model | Factor | CLHLS | | |
|---|---|---|---|---|
| | | **Crude** | **Age-sex adjusted** | **All covariates adjusted †** |
| Model D | Plasma albumin (g/L) | **0.95 (0.94, 0.96)**\*\*\* | **0.97 (0.96, 0.99)**\*\*\* | **0.97 (0.95, 0.99)**\*\*\* |
| Model E | Uric acid (umol/L) | 1.000 (0.999, 1.001) | 1.001 (1.000, 1.002) | 1.001 (1.000, 1.001) |
| Model F | Serum creatinine (μmol/L) | 1.002 (0.999, 1.004) | 1.002 (0.999, 1.004) | 1.002 (1.000, 1.005) |
| Model G | Albumin creatinine ratio (mg/g) | **1.000 (1.000, 1.000)**\* | 1.000 (1.000, 1.000) | 1.000 (1.000, 1.000) |
| Model H | Categorical ACR | | | |
| | <30 | Ref | Ref | Ref |
| | ≥30 | **1.30 (1.11, 1.53)**\*\* | **1.19 (1.02, 1.40)**\* | **1.21 (1.02, 1.43)**\* |
| Model I | eGFR | **0.99 (0.99, 0.996)**\*\*\* | 0.998 (0.99, 1.003) | 0.998 (0.99, 1.003) |
| Model J | Categorical eGFR | | | |
| | <30 | 1.55 (0.79, 3.04) | 0.85 (0.43, 1.69) | 0.98 (0.49, 1.98) |
| | 30~ | 1.39 (0.74, 2.59) | 0.80 (0.42, 1.51) | 0.82 (0.43, 1.55) |
| | 45~ | 1.08 (0.59, 1.98) | 0.64 (0.35, 1.19) | 0.61 (0.33, 1.14) |
| | 60~ | 0.98 (0.54, 1.78) | 0.65 (0.35, 1.19) | 0.67 (0.36, 1.23) |
| | 90~ | Ref | Ref | Ref |
| Model K | CKD | | | |
| | No | Ref | Ref | Ref |
| | Yes | **1.24 (1.07, 1.44)**\*\* | 1.09 (0.94, 1.26) | 1.04 (0.89, 1.21) |

Abbreviations: HR = Hazard ratio CI = confidence interval, eGFR = estimated glomerular filtration rate, CKD = chronic kidney diseases.

\*\*\* p<0.001

\*\*p<0.01

\*p<0.05.

† Adjusted for age, gender, race, educational level, income, marital status, health condition, smoking status, drinking status, physical activity, body mass index, hypertension and diabetes.

between CKD/eGFR and mortality was found stronger in the US sample. Notably, the biomarker level and these associations varied in different age groups.

## CKD-related biomarkers comparison between China and the US

Some prior studies compared the CKD-related biomarker levels between China and the US among adults aged≥20 years old. They found that the prevalence of albuminuria, defined as elevated ACR, was 8.1% in the US adults versus 9.5% in China, with the median of ACR as 5.97 mg/g in the US and 6.7 mg/g in China in 2009–2010 [5]. This finding was opposite to ours, which might be caused by the different age ranges of participants. However, another study comparing the prevalence of CKD found that the weighted mean of ACR was 15.67 mg/g in Chinese in 2006, while 22.81 mg/g, 53.44 mg/g, 32.89 mg/g in Whites, African Americans, and Hispanics, respectively, in 1999–2006 [6]. Interestingly, some also found that the ACR of Chinese adults in 2007–2010 and that of participants from the US in 2005–2010 were the same, about 6.3 mg/g [18]. As for serum creatinine, studies showed the same results as ours that the level in the US was higher than that in China, though all within the normal range [5, 18]. Previous studies comparing CKD between China and the US did not take BUN as an index. Moreover, some studies indicated that eGFR of Chinese was higher than that of US population, especially the Whites [5, 6]. Nevertheless, a study had the same finding as ours that eGFR in China was lower than that in the NHANES participants [18]. Although the mean

**Table 6. Hazard ratio (95% CI) of biomarkers on mortality in US participants (NHANES 2011–2014).**

| Model | Factor | NHANES | | |
|---|---|---|---|---|
| | | Crude | Age-sex adjusted | All covariates adjusted † |
| **Total** | | | | |
| Model A | Urinary albumin (mg/L) | **1.000 (1.000, 1.001)**\*** | **1.000 (1.000, 1.000)**\*** | **1.000 (1.000, 1.000)**\*** |
| Model B | Urinary creatinine (mg/dL) | 1.001 (0.999, 1.003) | 1.002 (0.999, 1.004) | 1.001 (0.998, 1.003) |
| Model C | Blood urea nitrogen (mmol/L) | **1.14 (1.12, 1.17)**\*** | **1.12 (1.09, 1.15)**\*** | **1.11 (1.07, 1.14)**\*** |
| Model D | Plasma albumin (g/L) | **0.87 (0.84, 0.90)**\*** | **0.88 (0.85, 0.92)**\*** | **0.89 (0.86, 0.93)**\*** |
| Model E | Uric acid (umol/L) | **1.003 (1.002, 1.005)**\*** | **1.003 (1.001, 1.004)**\*** | **1.003 (1.001, 1.004)**\** |
| Model F | Albumin creatinine ratio (mg/g) | **1.000 (1.000, 1.000)**\*** | **1.000 (1.000, 1.000)**\*** | **1.000 (1.000, 1.001)**\*** |
| Model G | Serum creatinine (μmol/L) | **1.005 (1.004, 1.006)**\*** | **1.004 (1.003, 1.006)**\*** | **1.004 (1.003, 1.006)**\*** |
| Model H | Categorical ACR | | | |
| | <30 | Ref | Ref | Ref |
| | ≥30 | **3.01 (2.28, 3.97)**\*** | **2.31 (1.74, 3.05)**\*** | **2.11 (1.55, 2.86)**\*** |
| Model I | eGFR | **0.97 (0.96, 0.98)**\*** | **0.98 (0.97, 0.99)**\*** | **0.98 (0.97, 0.99)**\*** |
| Model J | CKD | | | |
| | No | Ref | Ref | Ref |
| | Yes | **3.65 (2.68, 4.96)**\*** | **2.47 (1.80, 3.40)**\*** | **2.18 (1.56, 3.04)**\*** |
| Model K | Categorical eGFR | | | |
| | <30 | **10.09 (5.08, 20.04)**\*** | **4.59 (2.26, 9.32)**\*** | **3.56 (1.71, 7.42)**\*** |
| | 30~ | **6.37 (3.39, 11.97)**\*** | **2.40 (1.24, 4.66)**\** | **2.25 (1.14, 4.43)**\* |
| | 45~ | **3.41 (1.85, 6.29)**\*** | 1.46 (0.77, 2.77) | 1.44 (0.75, 2.76) |
| | 60~ | **1.90 (1.06, 3.42)**\* | 1.07 (0.59, 1.96) | 1.11 (0.60, 2.06) |
| | 90~ | Ref | Ref | Ref |
| **Age: 65–69** | | | | |
| Model A | Urinary albumin (mg/L) | **1.001 (1.000, 1.001)**\*** | **1.001 (1.000, 1.001)**\*** | **1.001 (1.000, 1.002)**\** |
| Model B | Urinary creatinine (mg/dL) | **1.005 (1.001, 1.01)**\* | **1.005 (1.000, 1.010)**\* | 1.004 (0.999, 1.010) |
| Model C | Blood urea nitrogen (mmol/L) | **1.12 (1.02, 1.236)**\* | **1.12 (1.01, 1.23)**\* | 1.08 (0.95, 1.23) |
| Model D | Plasma albumin (g/L) | **0.87 (0.80, 0.95)**\** | **0.87 (0.80, 0.95)**\** | **0.84 (0.74, 0.95)**\* |
| Model E | Uric acid (umol/L) | 1.002 (0.997, 1.007) | 1.001 (0.996, 1.01) | 1.002 (0.996, 1.010) |
| Model F | Albumin creatinine ratio (mg/g) | 1.000 (1.000, 1.001) | 1.000 (1.000, 1.000) | 1.000 (1.000, 1.001) |
| Model G | Serum creatinine (μmol/L) | **1.006 (1.001, 1.01)**\** | **1.005 (1.001, 1.01)**\* | 1.004 (0.998, 1.010) |
| Model H | Categorical ACR | | | |
| | <30 | Ref | Ref | Ref |
| | ≥30 | **3.17 (1.31, 7.64)**\* | **3.07 (1.27, 7.44)**\* | **3.49 (1.23, 9.91)**\* |
| Model I | eGFR | **0.97 (0.96, 0.99)**\** | **0.97 (0.96, 0.99)**\** | **0.98 (0.96, 0.998)**\* |
| Model J | CKD | | | |
| | No | Ref | Ref | Ref |
| | Yes | **3.64 (1.54, 8.65)**\** | **3.59 (1.51, 8.54)**\** | **4.10 (1.39, 12.08)**\* |
| Model K | Categorical eGFR | | | |
| | <30 | **6.85 (1.25, 37.39)**\* | **6.64 (1.20, 36.85)**\* | 5.15 (0.72, 36.82) |
| | 30~ | 5.07 (0.93, 27.73) | 5.04 (0.90, 28.12) | **12.09 (1.55, 94.16)**\* |
| | 45~ | **3.85 (1.04, 14.38)**\* | 3.73 (0.99, 13.98) | **5.94 (1.00, 35.33)**\* |
| | 60~ | 1.20 (0.36, 3.98) | 1.16 (0.35, 3.88) | 2.21 (0.57, 8.53) |
| | 90~ | Ref | Ref | Ref |
| **Age: 70–74** | | | | |
| Model A | Urinary albumin (mg/L) | **1.001 (1.000, 1.002)**\* | **1.001 (1.000, 1.002)**\* | **1.001 (1.000, 1.002)**\* |
| Model B | Urinary creatinine (mg/dL) | 1.002 (0.996, 1.01) | 1.002 (0.995, 1.01) | 1.001 (0.995, 1.01) |
| Model C | Blood urea nitrogen (mmol/L) | 0.98 (0.81, 1.18) | 0.97 (0.80, 1.17) | 0.95 (0.77, 1.18) |

*(Continued)*

**Table 6.** (Continued)

| Model | Factor | NHANES | | |
|---|---|---|---|---|
| | | Crude | Age-sex adjusted | All covariates adjusted † |
| Model D | Plasma albumin (g/L) | **0.84 (0.76, 0.94)**\*\* | **0.84 (0.75, 0.93)**\*\* | **0.87 (0.77, 0.98)**\* |
| Model E | Uric acid (umol/L) | 0.999 (0.995, 1.004) | 0.999 (0.994, 1.004) | 0.999 (0.994, 1.004) |
| Model F | Albumin creatinine ratio (mg/g) | 1.001 (1.000, 1.002) | 1.001 (1.000, 1.002) | 1.001 (1.000, 1.003) |
| Model G | Serum creatinine (μmol/L) | 1.00 (0.99, 1.01) | 1.00 (0.99, 1.01) | 1.00 (0.98, 1.02) |
| Model H | Categorical ACR | | | |
| | <30 | Ref | Ref | Ref |
| | ≥30 | **2.65 (1.24, 5.67)**\* | **2.60 (1.22, 5.58)**\* | 2.13 (0.88, 5.15) |
| Model I | eGFR | 1.00 (0.98, 1.02) | 1.00 (0.98, 1.03) | 1.01 (0.98, 1.03) |
| Model J | CKD | | | |
| | No | Ref | Ref | Ref |
| | Yes | **2.23 (1.06, 4.69)**\* | **2.18 (1.03, 4.60)**\* | 1.79 (0.76, 4.20) |
| Model K | Categorical eGFR | | | |
| | <30 | 2.90 (0.34, 25.02) | 2.65 (0.30, 23.31) | **20.27 (1.30, 316.77)**\* |
| | 30~ | NA | NA | NA |
| | 45~ | 1.54 (0.49, 4.89) | 1.46 (0.46, 4.67) | 0.94 (0.26, 3.44) |
| | 60~ | 0.85 (0.31, 2.35) | 0.82 (0.29, 2.27) | 0.67 (0.22, 2.01) |
| | 90~ | Ref | Ref | Ref |
| **Age: 75–79** | | | | |
| Model A | Urinary albumin (mg/L) | **1.000 (1.000, 1.001)**\*\*\* | **1.001 (1.000, 1.001)**\*\*\* | **1.001 (1.000, 1.001)**\*\*\* |
| Model B | Urinary creatinine (mg/dL) | 1.002 (0.997, 1.01) | 1.001 (0.996, 1.006) | 0.999 (0.99, 1.01) |
| Model C | Blood urea nitrogen (mmol/L) | **1.11 (1.06, 1.17)**\*\*\* | **1.11 (1.05, 1.16)**\*\*\* | **1.12 (1.04, 1.20)**\*\* |
| Model D | Plasma albumin (g/L) | **0.79 (0.72, 0.87)**\*\*\* | **0.80 (0.73, 0.87)**\*\*\* | **0.80 (0.71, 0.90)**\*\*\* |
| Model E | Uric acid (umol/L) | **1.005 (1.002, 1.008)**\*\* | **1.005 (1.001, 1.008)**\*\* | **1.004 (1.000, 1.009)**\* |
| Model F | Albumin creatinine ratio (mg/g) | **1.001 (1.000, 1.001)**\*\*\* | **1.001 (1.000, 1.001)**\*\*\* | **1.001 (1.000, 1.001)**\*\*\* |
| Model G | Serum creatinine (μmol/L) | **1.004 (1.002, 1.006)**\*\*\* | **1.004 (1.002, 1.006)**\*\*\* | **1.004 (1.002, 1.007)**\*\* |
| Model H | Categorical ACR | | | |
| | <30 | Ref | Ref | Ref |
| | ≥30 | **4.00 (1.95, 8.20)**\*\*\* | **4.07 (2.00, 8.44)**\*\*\* | **5.25 (2.23, 12.37)**\*\*\* |
| Model I | eGFR | **0.97 (0.95, 0.99)**\*\*\* | **0.97 (0.95, 0.99)**\*\*\* | **0.96 (0.94, 0.98)**\*\*\* |
| Model J | CKD | | | |
| | No | Ref | Ref | Ref |
| | Yes | **2.75 (1.23, 6.14)**\* | **2.88 (1.29, 6.47)**\* | **2.95 (1.25, 6.97)**\* |
| Model K | Categorical eGFR | | | |
| | <30 | **10.51 (1.29, 85.59)**\* | **11.22 (1.37, 91.68)**\* | **20.21 (1.86, 219.37)**\* |
| | 30~ | 2.38 (0.26, 21.38) | 2.55 (0.28, 22.94) | 4.41 (0.41, 47.68) |
| | 45~ | 1.25 (0.15, 10.75) | 1.27 (0.15, 10.95) | 1.37 (0.13, 14.47) |
| | 60~ | 1.54 (0.20, 11.73) | 1.52 (0.20, 11.63) | 1.91 (0.20, 18.10) |
| | 90~ | Ref | Ref | Ref |
| **Age: 80+** | | | | |
| Model A | Urinary albumin (mg/L) | 1.000 (1.000, 1.000) | 1.000 (1.000, 1.000) | 1.000 (1.000, 1.001) |
| Model B | Urinary creatinine (mg/dL) | 1.002 (0.999, 1.005) | 1.001 (0.998, 1.004) | 1.000 (0.996, 1.003) |
| Model C | Blood urea nitrogen (mmol/L) | **1.13 (1.09, 1.18)**\*\*\* | **1.14 (1.09, 1.19)**\*\*\* | **1.13 (1.08, 1.19)**\*\*\* |
| Model D | Plasma albumin (g/L) | **0.93 (0.88, 0.99)**\* | **0.93 (0.88, 0.98)**\* | 0.95 (0.89, 1.01) |
| Model E | Uric acid (umol/L) | **1.004 (1.002, 1.006)**\*\*\* | **1.003 (1.002, 1.005)**\*\*\* | **1.004 (1.002, 1.006)**\*\*\* |
| Model F | Albumin creatinine ratio (mg/g) | **1.000 (1.000, 1.000)**\* | **1.000 (1.000, 1.000)**\* | **1.000 (1.000, 1.001)**\*\* |
| Model G | Serum creatinine (μmol/L) | **1.007 (1.004, 1.009)**\*\*\* | **1.006 (1.004, 1.009)**\*\*\* | **1.007 (1.004, 1.01)**\*\*\* |

(*Continued*)

**Table 6.** (Continued)

| Model | Factor | NHANES | | |
|---|---|---|---|---|
| | | Crude | Age-sex adjusted | All covariates adjusted † |
| Model H | Categorical ACR | | | |
| | <30 | Ref | Ref | Ref |
| | ≥30 | **1.80 (1.26, 2.57)**\*\* | **1.79 (1.26, 2.56)**\*\* | **1.85 (1.24, 2.77)**\*\* |
| Model I | eGFR | **0.98 (0.97, 0.99)**\*\*\* | **0.98 (0.97, 0.99)**\*\*\* | **0.98 (0.97, 0.99)**\*\*\* |
| Model J | CKD | | | |
| | No | Ref | Ref | Ref |
| | Yes | **2.15 (1.39, 3.31)**\*\*\* | **2.15 (1.39, 3.32)**\*\*\* | **2.06 (1.29, 3.29)**\*\* |
| Model K | Categorical eGFR | | | |
| | <30 | 3.49 (0.98, 12.37) | 3.95 (1.11, 14.04)\* | 3.37 (0.86, 13.21) |
| | 30~ | 2.56 (0.78, 8.36) | 2.54 (0.78, 8.30) | 2.45 (0.68, 8.83) |
| | 45~ | 1.38 (0.42, 4.52) | 1.38 (0.42, 4.50) | 1.43 (0.40, 5.12) |
| | 60~ | 1.10 (0.34, 3.55) | 1.11 (0.34, 3.58) | 1.21 (0.34, 4.35) |
| | 90~ | Ref | Ref | Ref |

Abbreviations: HR = Hazard ratio CI = confidence interval, eGFR = estimated glomerular filtration rate, CKD = chronic kidney diseases.

\*\*\* $p < 0.001$

\*\*$p < 0.01$

\*$p < 0.05$.

† Adjusted for age, gender, race, educational level, income, marital status, health condition, smoking status, drinking status, physical activity, body mass index, hypertension and diabetes.

eGFR in our study was lower in in the Chinese sample, it may result from the larger proportion of participants aged 80 and above in CLHLS than NHANES. The age-specific eGFR was higher in the Chinese participants, except for the eldest group.

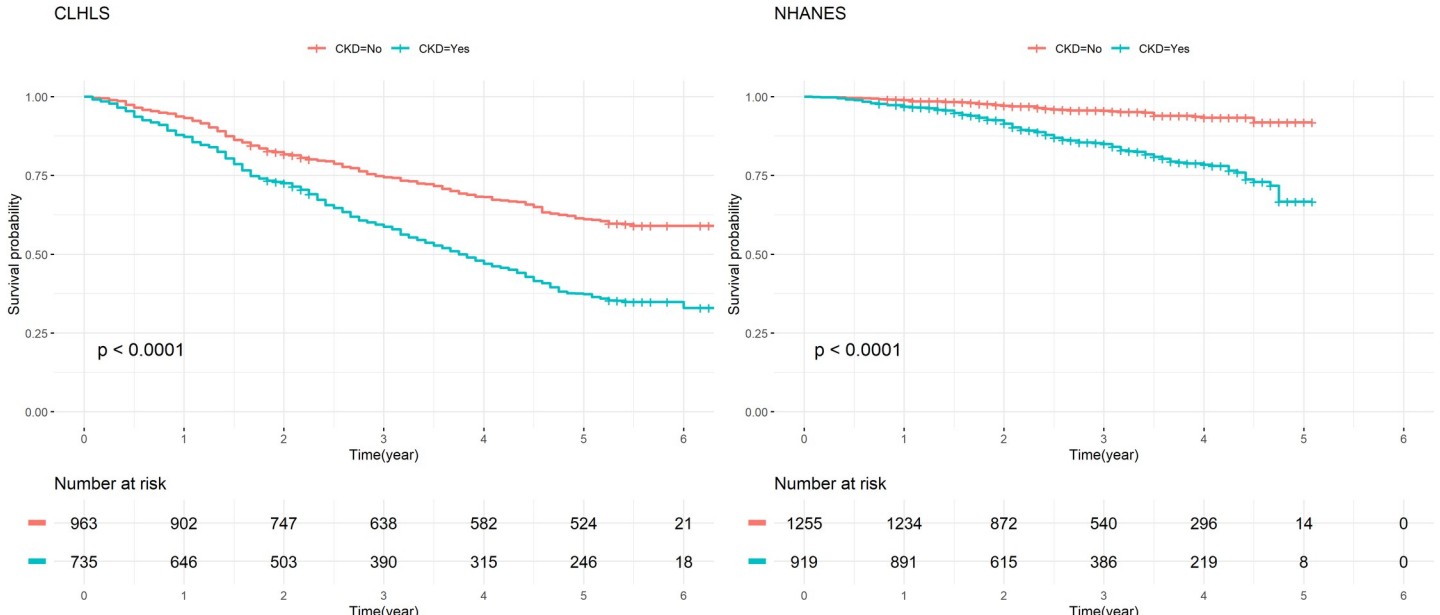

**Fig 1. Kaplan-Meier Curve of CKD.** CLHLS: 2012–2018. NHANES: 2011/2013-2015.

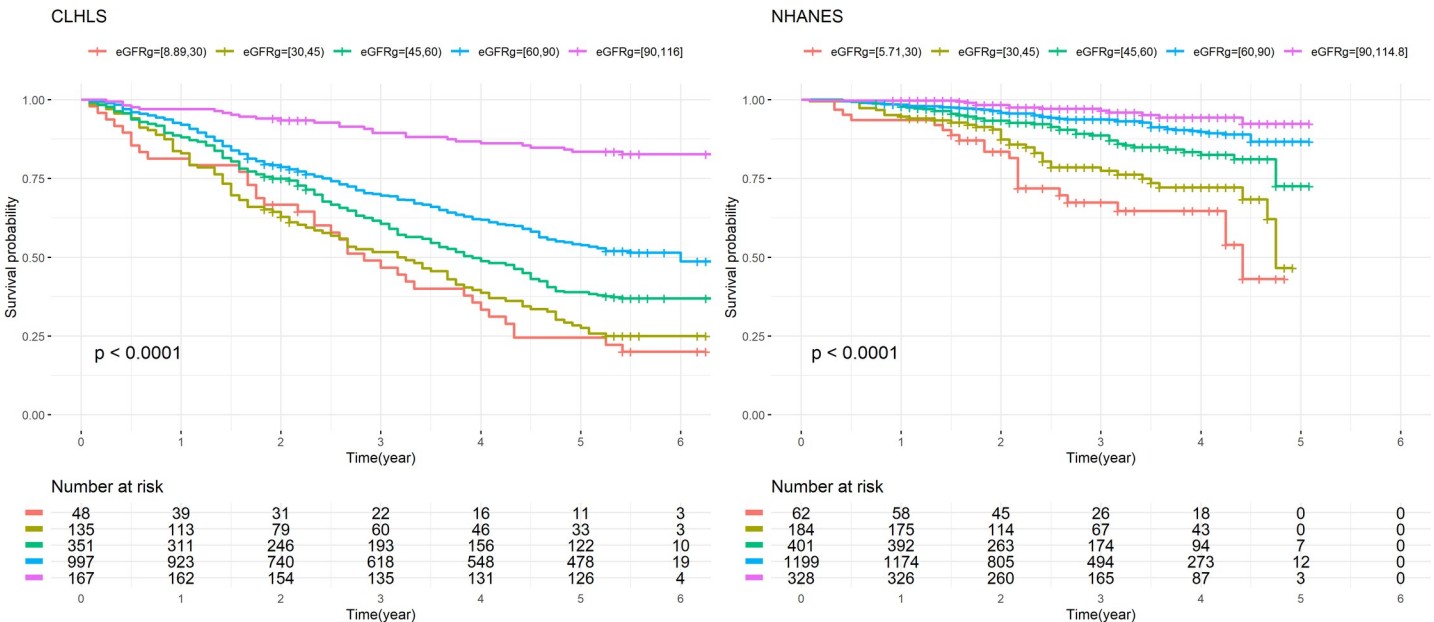

**Fig 2. Kaplan-Meier Curve of CKD stage.** CLHLS: 2012–2018. NHANES: 2011/2013-2015.

## Risk factors for CKD

In a previous study, it was suggested that CKD was associated with increased age, hypertension, diabetes, cardiovascular diseases, and hyperuricemia in both Chinese and the US population [5]. This study also suggested that being female was associated with decreased eGFR in CLHLS participants but not in the NHANES sample, which is consistent with our results. Besides, central obesity was also indicated associated with CKD. Another study illustrated that diabetes is more closely related to CKD in Whites, African Americans, and Hispanics, with Chinese as reference, and overweight was less associated with CKD in Whites [6]. We additionally adjusted for more variables like household income than these previous studies. The weaker relationship between higher BMI and CKD may be reasonable considering the huge gap between the proportion of overweight people in the Chinese and the US samples. Furthermore, contrary to the positive health effect of physical activity in general, we found no physical activity was associated to lower risk of having CKD among Chinese participants. We speculated that the presence of CKD might be more related to age than to physical activity. It is possible that doing exercise enable people to live longer but, thus, have CKD. This may also explain why never smokers among Chinese elderly showed a higher prevalence of CKD.

## CKD and mortality risk

A previous comprehensive meta-analysis established the association of reduced eGFR with all-cause mortality in the general population [19]. Our findings are consistent with the previous result that all-cause mortality increased at eGFRs lower than 60 mL/min/1.73 m$^2$. The meta-analysis showed that the adjusted HRs for all-cause mortality were 1.57 (95% CI: 1.39–1.78) for eGFR 45 mL/min/1.73 m$^2$, and 3.14 (2.39–4.13) for eGFR 15 mL/min/1.73 m$^2$, which was similar to the HRs of NHANES group in our study. As for the different effect size between the two populations, a study found that the linear association between eGFR and all-cause mortality appeared clearer in US general population compared with the Chinese population, and the association was insignificant even in the lowest eGFR spline, which was potentially caused by

the limited death events or sample size [18]. Our findings present a similar disparity of effect size. Another study suggested that the relative mortality of lower eGFR was largely similar among Asians, whites, and blacks [20].

## Possible reasons for the differences between the China sample and US sample

There are a number of possible reasons for the racial disparities in CKD. First, African Americans might have a higher prevalence of poverty compared to whites, and low socioeconomic status (SES) was found to have a stronger association with CKD among African Americans than among whites. It was indicated that the impact of SES may lead to the racial differences in biology [21]. Second, higher prevalences of comorbidities and obesity in the US population could explain their mortality rate of more than double that in China [18]. In a retrospective population-based cohort study of 530,771 adults with CKD residing in Alberta, Canada between 2003 and 2011, it was found that a number of comorbidities could increase the risk of hospitalization, including not only hypertension and diabetes, but also mental health, chronic pain, dementia and cancer [22]. By contrast, the most common risk factor of CKD in China was chronic glomerulonephritis [23], and nontraditional risk factors such as fetal and maternal factors, infections, environmental factors, and acute kidney injury were also major threats [24]. These risk factors may be linked to social deprivation and poverty in developing countries, working directly through the accessibility of predisposition, diagnosis, and management or indirectly through the increased health care burden [8]. Therefore, we assumed the elder women in China might have a high rate of reproductive tract infection several decades ago, which might have led to CKD later, as poor personal hygiene and low living standard at that time. However, we did not examine the comprehensive comorbidity condition including chronic glomerulonephritis among participants, so further studies may need to explore the gap of comorbidities in the Chinese elderly and those in the US. Apart from that, a study of patients on dialysis found a higher prevalence of cardiovascular disease at the start of dialysis in whites compared with other racial groups. Because atherosclerosis is a common and significant cause of morbidity and mortality for patients with end-stage renal disease, it is understandable that whites with more atherosclerosis might have a higher risk of death with CKD, especially in the end-stage [25]. Moreover, the cumulative dose of cigarette smoking and additional risk factors in various racial groups are also possible to result in the survival disparities of patients with end-stage renal disease [25]. In addition, the difference of the findings between the two countries varied among different age groups. There were a much larger proportion of the population aged 80 or older in the China sample than the US sample (65.5% vs. 26.1%). There may be survivor bias caused by a relatively healthier Chinese population.

## Study strengths and limitations

Our study has several strengths. First, the cohort size of CLHLS and NHANES was quite large, and represented an older population which was less studied before. Second, the survey methods of CLHLS and NHANES were appropriate and time-tested. We used the data of recent waves which could best estimate the current situation about CKD. Additionally, we took a diverse group of variables to assess. Besides, we applied the same definition of CKD (original CKD-EPI creatinine equation) which have been validated in both populations, and proved more accurate than other equations, allowing eGFR levels between the two populations to be comparable.

However, our study had some limitations as well. First, CLHLS sample weight is calculated based on the total cohort and not the biomarker cohort, and only considered the age–sex–

urban/rural residence-specific distribution of the population. A previous CLHLS study suggested not including weight in multivariate analyses since the weight does not capture other important compositional variables like economic and education status and weighted regressions will unnecessarily increase standard errors [26]. We excluded sample weights from CLHLS for this reason and from NHANES for consistency, and we present results incorporating the sample weights for both studies in S7–1–S9 Tables. Lack of appropriate weight data limited the representativeness to the general population; CLHLS oversampled rural residents in China while NHANES oversampled minorities and lower SES individuals in the US. Second, the measuring technique for biomarkers may differ in NHANES and CLHLS, which made the value of biomarkers less comparable. For example, serum creatinine in NHANES was standardized to IDMS while it was measured by the picric acid method in CLHLS. Thirdly, the definition of the covariates in NHANES and CLHLS were not exactly the same due to the different questionnaire in surveys. The heterogeneity in variable definition did not permit pooling of data and evaluation of an interaction. Moreover, the age distributions were different across the two populations. Most Chinese participants aged 80 or older. Therefore, we also reported the age stratified analyses and identified some difference across the age groups. This may explain part of the difference between the two populations. Last but not least, there was a much higher loss to follow-up in the CLHLS sample, compared to that in NHANES. This might decrease the credibility of the mortality results in CLHLS and make the results of two countries less comparable to a small extent.

## Conclusion

In conclusion, the elderly population in the NHANES have worse CKD-related biomarker levels than in CLHLS, and the factors associated with CKD from demo-social to lifestyle factors vary in the two cohorts. Moreover, the mortality rate from CKD and the association between CKD and mortality was higher in the NHANES than in CLHLS. Further studies are warranted to validate our findings and elucidate the biological mechanisms.

## Supporting information

**S1 Table. Cohorts' characteristics of this study.**
(PDF)

**S2 Table.** 1. Characteristics of the included and excluded participants in CLHLS. 2. Characteristics of the included and excluded participants in NHANES.
(ZIP)

**S3 Table. Equations used to estimate glomerular filtration rate eGFR (CKD-EPI: Chronic Kidney Disease Epidemiology Collaboration).**
(PDF)

**S4 Table.** 1. Demographic characteristics and median (P25-P75) of biomarkers (Chinese participants: CLHLS 2012). 2. Demographic characteristics and median (P25-P75) of biomarkers (US participants: NHANES 2011–2014).
(ZIP)

**S5 Table. Odds ratio (95% CI) of factors associated with abnormal eGFR (<60) in CLHLS and NHANES.**
(PDF)

**S6 Table. Odds ratio (95% CI) of factors associated with abnormal ACR ($\geq$30 mg/g) in CLHLS and NHANES.**
(PDF)

**S7 Table.** 1. Demographic characteristics and weighted mean (SD) of biomarkers (Chinese participants: CLHLS 2012). 2. Demographic characteristics and weighted mean (SD) of biomarkers (US participants: NHANES 2011–2014).
(ZIP)

**S8 Table. Odds ratio (95% CI) of factors associated with CKD in Chinese and US population (weighted).**
(PDF)

**S9 Table. Hazard ratio (95% CI) of CKD biomarkers on mortality in Chinese and US population (weighted).**
(PDF)

## Acknowledgments

We thank Daqing Guo, MD, FASN for his clinical input.

## Author Contributions

**Conceptualization:** Yeli Wang, Tazeen Hasan Jafar.

**Data curation:** Hui Miao, Linxin Liu.

**Formal analysis:** Hui Miao, Linxin Liu, Yucheng Wang, Qile He.

**Funding acquisition:** John S. Ji.

**Methodology:** Hui Miao, Linxin Liu, Yeli Wang, Tazeen Hasan Jafar, John S. Ji.

**Project administration:** John S. Ji.

**Resources:** Yi Zeng.

**Supervision:** Yeli Wang, John S. Ji.

**Visualization:** Linxin Liu.

**Writing – original draft:** Hui Miao, Linxin Liu, Yucheng Wang, Qile He.

**Writing – review & editing:** Hui Miao, Linxin Liu, Yeli Wang, Tazeen Hasan Jafar, Shenglan Tang, Yi Zeng, John S. Ji.

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
