## [Decision Letter · Decision Letter 0]

20 May 2021

PONE-D-21-07064

Conceptualization semicolons

A Comparison Study of Chronic Kidney Disease Biomarkers and Mortality among Older Adults in China and the United States

PLOS ONE

Dear Dr. Ji,

Thank you for submitting your manuscript to PLOS ONE. After careful consideration, we feel that it has merit but does not fully meet PLOS ONE’s publication criteria as it currently stands. Therefore, we invite you to submit a revised version of the manuscript that addresses the points raised during the review process.

The manuscript presents interesting data and is of potential interest given the scale of the analysis. However, there are several important limitations that preclude to accept the manuscript in its current form. We invite you to submit a revised version of the manuscript that addresses the points below:

Please keep special attention to the Reviewers' suggestions, including:

1. Need to perform analysis with inclusion of the survey weights.

2. Need to report results for each age groups or age-standardization.

3. Better clarification of differences between populations considering age-specific eGFR.

4. Inclusion of additional factors in the model.

5. Enhancement of the "Discussion" section.

Additionally, I suggest to:

1. Rephrase some sentences (for example, the first phrase of the "Introduction" is not fully compliant with the KDIGO definition, the phrase "In CLHLS, the next of kin reported the mortality information in the follow-up surveys in 2014 and 2017/2018." should be changed for clarity) and edit the text in some places for better connection between concepts and paragraphs (the second paragraph of the "Introduction" starting with "China is a middle-income..." would be better to initiate with something like "Previous studies revealed substantial difference between China and USA in the ...").

2. Indicate more details for the Chinese Longitudinal Healthy Longevity Study, and provide after the phrase "The study design and sampling method were described previously (8)" some key information with "In brief, ...". These additional details could better explain the reasons of differences between CLHLS and NHANES. Probably, a table with some core characteristics of both surveys would allow to present them in a clear and concise way.

3. Clearly indicate in the "Methods" whether the determination of serum creatinine measurement procedures has been standardized to isotope dilution mass spectrometry (IDMS).

4. Provide in the "Data sharing statement" information about the data availability (not "additional" but the surveys data).

5. Please use "urinary albumin" instead of "urinary microalbumin".

6. Use IQR instead of range for the description of ACR and other biomarkers in the manuscript.

7. Instead of "Predictors of CKD" that could be revealed in the longitudinal study, the cross-sectional part of the current analysis (related to Table 2) should use the terminology "Factors associated with CKD".

8. The data provided in the "Figure 1" is widely available, and should be moved to the Supplementary.

9. Change legends in figures: instead of "CKD=1 / CKD=0" use "CKD / no CKD", don't use "Strata" in the legend.

10. As you correctly identified in the "Methods", the KDIGO defines CKD based on eGFR and albuminuria. The current analysis merge persons with either low eGFR or increased albuminuria into a single "CKD" group, and presents the data according to presence/absence of CKD. It would be a great advantage if, in addition to this already available analysis, the authors will present a separate analysis for "normal vs low eGFR" and "normal vs increased albuminuria" groups.

11. Please review the tables' titles for clarity (both in the main manuscript and Supplement).

12. Please unify the data representation in tables. In the Supplementary table 2 some cells have no decimal signs, some have (or there is some mistyping): for example in the ACR column for different age groups the values are "8.4 (5.4,17.9)", "10.3 (6,22.7)", etc.

13. Please clarify in "Methods" the phrase "In the logistic regression and the Cox models, missing value of covariates were not excluded, but coded as a categorical variable."

We look forward to receiving your revised manuscript.

Kind regards,

Boris Bikbov

Academic Editor

PLOS ONE

Journal Requirements:

2.Thank you for providing consent information in the ethics statement of the online submission form. Please also state whether written informed consent was obtained from participants of the CLHLS study.

3. Thank you for providing the date(s) when patient medical information was initially recorded. Please also include the date(s) on which your research team accessed the databases/records to obtain the retrospective data used in your study.

4. Please provide references for the NHANES and CLHLS studies if any exist.

5. Please include your actual numerical p-values in Tables 3, S3, S6, S8, and S9.

6.We note that you have indicated that data from this study are available upon request. PLOS only allows data to be available upon request if there are legal or ethical restrictions on sharing data publicly. For information on unacceptable data access restrictions, please see http://journals.plos.org/plosone/s/data-availability#loc-unacceptable-data-access-restrictions.

7. Please amend either the title on the online submission form (via Edit Submission) or the title in the manuscript so that they are identical.

Reviewers' comments:

Reviewer's Responses to Questions

**Comments to the Author**

1. Is the manuscript technically sound, and do the data support the conclusions?

Reviewer #1: No

Reviewer #2: Yes

2. Has the statistical analysis been performed appropriately and rigorously? 

Reviewer #1: No

Reviewer #2: Yes

3. Have the authors made all data underlying the findings in their manuscript fully available?

Reviewer #1: Yes

Reviewer #2: Yes

4. Is the manuscript presented in an intelligible fashion and written in standard English?

Reviewer #1: Yes

Reviewer #2: Yes

5. Review Comments to the Author

Reviewer #1: 1. Summary

The study summarizes chronic kidney disease biomarkers and mortality among elderly participants in the NHANES survey in the United States and the CLHLS survey in China. It reports on many biomarkers, highlighting a few where they identify differences. The article also reports differences in the mortality among CKD patients in the two surveys and briefly hypothesizes some possible reasons for the differences. However, the authors do not discuss the implications of the overall findings (especially where inconsistencies were found) but rather they summarize tabulated data. More importantly, they overstate the generalizability of the results because they did not incorporate the survey weights into their analyses. This last point is the primary weakness of the study and makes it impossible to accurately assess differences between the two populations. I recommend major revisions before acceptance.

2. Areas for improvement

Major issues:

The authors should repeat the analysis with inclusion of the survey weights of each survey. This current omission prevents comparison and generalizability to the populations these surveys represent. It was briefly noted in the Discussion (lines 474-477) that weights were not used, with no discussion of why or what the implications are. This was a few sentences after the cohort size and representativeness of the samples was incorrectly touted as a strength of the study (lines 467-477). The conclusion (lines 484-487) generalizes to the countries by stating differences were found between “the elderly population in the US” and “in China”, which the study has failed to show.

Also, as the age distributions of the survey participants are quite different between the two samples (lines 246-247), comparison should occur within detailed age groups, or age-standardized across included age groups. This will more accurately highlight real differences in biomarkers or mortality that are not simply due to ageing. For example, reported prevalence of CKD in line 264 could also be reported as age-standardized to better compare the two samples. The age breakdown in lines 267-270 is helpful and better conveys the differences (higher rates in all age groups in US sample than in China sample) than the overall rates in line 264 (higher overall rate in China sample), which is misleading. This is also an issue in the discussion, where line 410 claims that eGFR is lower in the China sample than in NHANES, but line 413 acknowledges that the age-specific eGFR was higher in the Chinese sample for all but the oldest age group. The authors should take care to report and interpret meaningful results.

The conclusion of the abstract supposes that some of the differences may be due to co-morbid conditions (line 53). But this would easily be partially accounted for by at least including indicators for diabetes and hypertension in the adjusted model. Calculating separate results among those with diabetes and/or hypertension is coarse and not very useful, as the authors themselves note that diabetes and hypertension prevalence differs quite a bit between the study populations. Given the known associations between these two conditions and CKD, they should be included in the adjusted models.

In the discussion, differences were repeated but the “why” was not explored. For example, lines 389-394 report that females had higher levels of CKD in the Chinese sample, but there was no association between sex and CKD diagnosis in the US sample. But there was no discussion about why this might be observed. Also, in the US, high household income and low education level were associated with CKD prevalence (lines 392-393) but these two characteristics are typically at odds with each other; usually higher income households have more education. I wish this apparent paradox had been somewhat explored or explained.

Minor issues:

GBD 2019 has been published for a while now and should replace the GBD 2017 source.

What are the implications of the differences in CKD biomarker measurement techniques? This is vaguely included as a limitation (lines 477-478) but not discussed at all in terms of how it affects interpretation of results. The title of the article should be revised to place less emphasis on biomarkers if the authors are not confident about the comparability of biomarker measurement between the samples or able to discuss the potential bias in more detail.

Baseline characteristics of % diabetes and % hypertension among participants should be reported.

The reported in-text results were cumbersome to read and difficult to draw conclusions from as a reader, but I acknowledge that can be difficult when reporting many different variables. (examples: lines 359-360 in particular, and paragraphs that contain lists of CKD prevalences)

It is unclear why the authors also conducted the analysis using the MDRD equation for the NHANES sample. Written justification would have strengthened the decision to include it.

Positive correlations should be assessed for both statistical significance and clinical relevance. The reported OR of 1.000 (1.000, 1.000) in line 351 is apparently statistically significant (with more decimal places?) but is meaningless clinically.

Edit for clarity:

- “bad at self-rated” (line 319)

- “associated with … White, other races…” (line 321)

- “In previous studies…” should be “In a previous study…” (line 416)

A study cited in the discussion found male sex predictive of CKD in a Chinese sample, which the authors claim “is consistent with our results (line 419). But line 389 states that female sex is predictive of CKD in this study’s Chinese sample, which is the opposite of the cited study.

I’m curious if the comparison studies described in the Discussion adjusted their models with the same or similar list of variables, or if differences in variables could account for some of the different results. (for example the study in lines 436-437 that reported different results than this study)

Was prevalence of chronic glomerulonephritis available in either of the surveys? If so, it should have been included in the adjusted models. If not, it would have been helpful to have it stated explicitly (since diabetes and hypertension were available). The one mention of this common primary renal diagnosis, especially in China, was insufficient given its potential importance to some of the differences shown. In renal registries, for example, survival among patients on renal replacement therapy is sometimes reported by primary renal diagnosis (especially diabetes, hypertension, chronic glomerulonephritis, and perhaps other causes), which would have supported the authors’ assertion that underlying comorbidities may account for differences in mortality. I believe from registries, that mortality is higher among patients with ESRD due to diabetes than those with ESRD due to chronic glomerulonephritis. So if more US patients have diabetes than the China samples, and vice versa with chronic glomerulonephritis, then this would be an easy connection to make to support the results.

I wasn’t clear by the end of the article exactly what the purpose of the study was or what conclusions I should draw from it are. The results can certainly be useful, but I wish the Introduction and Discussion had built up more motivation for and implications of the study.

Figures:

- Figure 1 should be a table.

- Figures 2 and 3 should be clearly labeled with CLHLS and NHANES.

Reviewer #2: Thanks for sharing this interesting work. Differences between populations are always a reason for hypothesis for future research regarding their causality (factors, genetic, racial, dietary or others). Very extensive information has been collected and compared, which is difficult to show clearly and is not at times overwhelming and confusing in its presentation.

The study hypothesis is clearly stated, and the population used is adequate.

The methodology is adequate, using population surveys. A problem that will creep into all the results analyzed is the age difference between the two populations. Although multivariate analyzes have been carried out trying to minimize the effect of the age difference between the populations, the use of a matched analysis with more similar populations could have been a useful strategy.

It looks like a comment at least should be done regarding the BMI classification for older adults (aged 65 or older) since a metanalysis published in 2014 by Winter and Al, has shown a different classification for this population (Winter JE, MacInnis RJ, Wattanapenpaiboon N, Nowson CA. BMI and all-cause mortality in older adults: a meta-analysis. Am J Clin Nutr. 2014 Apr;99(4):875-90) suggesting that optimal BMI for those >65 years is likely between 23-30, instead of 18.5 to 25.

Regarding the Results

The two populations studied differ widely in the age of the population, with the Chinese population averaging 13 years older (85 vs. 73 years).

This difference can explain many of the other differences found, even some that seem obvious like practicing physical activity. The fact that the Chinese population is much older than the US population should be discussed, it could be a survivor bias, being a "healthier" population.

In relation to the way of presenting the results, which involve a wealth of information, their presentation would be clearer and easier to read if additional headings were incorporated for the different results to be described., As well as for including headings at the discussion.

A minor commentary: the subtitle of the prevalence chapter would be more adequately described as “Difference in CKD prevalence between the two countries” or “Prevalence of CKD in both countries”.

An additional issue for the mortality analysis may be the loss to follow-up of 321 CLHLS participants (page 13 , line 128, representing 14% of the population. It should be investigated whether these participants were not lost to follow-up due to death, or comment that this was ruled out.

Regarding the limitations of the study, it should be mentioned the differences in the age of both population, that could itself explain many differences between both populations.

6. PLOS authors have the option to publish the peer review history of their article (what does this mean?). If published, this will include your full peer review and any attached files.

Reviewer #1: **Yes: **Sarah Wulf Hanson

Reviewer #2: **Yes: **Laura Sola

---

## [Author Response · Author response to Decision Letter 0]

13 Jul 2021

Comments from Academic Editor

1. Need to perform analysis with inclusion of the survey weights.

Response: We thank the editor for this suggestion. We want to let the editor know that this created twice the computational workload for our team, nevertheless, we performed additional analyses with weights and included them as supplemental tables. A couple of caveats, since there were only weights for the overall cohort instead of the subgroup who got biomarkers measured in CLHLS, the weighted results may not accurately reflect the situation of older people in China. We presented these analyses as supplemental tables, and did not interpret them in the manuscript for consistence and not to confuse the reader. We let the reader access them if needed. We want to note that the previous CLHLS study did not include sample weight in multivariate analysis because results from unweighted regression models produce unbiased coefficients when including variables related to sample selection (i.e., age, sex, and urbanicity) (Winship & Radbill, 1994) and that weighted regressions will unnecessarily increase standard errors.

2. Need to report results for each age groups or age-standardization.

Response: We agreed that the age difference should be further explored. Therefore, we reported the biomarkers level in different age groups in text (line 301-306). We also added age stratified analyses for CKD risk factor table and mortality risk tables (Table 2-5) and the corresponding description.

3. Better clarification of differences between populations considering age-specific eGFR.

Response: We saw an association between age and eGFR. Generally, kidney functions decreases with age, as expected. We saw this in both NHANES and CLHLS populations. 

We compared the eGFR in each age group between CLHLS and NHANES, and the difference varied in different age groups. eGFR were higher in China than the US in population aged younger than 80 and this difference almost disappeared in population aged 80 and older. We added the description in line 303-306.

4. Inclusion of additional factors in the model.

Response: We clarified adjustment variables in our models. We additionally adjusted for hypertension and diabetes in the model as the reviewer suggested. The results were similar before and after this adjustment. 

5. Enhancement of the "Discussion" section.

Response: We appreciate the comment on enhancing our discussion section. We did so by incorporating the many reviewer comments. 

Additional comments

1. Rephrase some sentences (for example, the first phrase of the "Introduction" is not fully compliant with the KDIGO definition, the phrase "In CLHLS, the next of kin reported the mortality information in the follow-up surveys in 2014 and 2017/2018." should be changed for clarity) and edit the text in some places for better connection between concepts and paragraphs (the second paragraph of the "Introduction" starting with "China is a middle-income..." would be better to initiate with something like "Previous studies revealed substantial difference between China and USA in the ...").

Response: We thank the editor for this helpful comment. We made the following edits in the Introduction section and the Methods section: 

“Chronic kidney disease (CKD) is a public health problem around the world. It refers to kidney damage or lasting low glomerular filtration rate (GFR), regardless of causes.” (Lines 64-65)

…

Previous studies revealed substantial difference between China and USA in the prevalence and mortality of CKD. (Lines 75-76)

…

“In CLHLS, the immediate family members of subjects reported the mortality information in the follow-up surveys in 2014 and 2017/2018. (Lines 190-191)”

2. Indicate more details for the Chinese Longitudinal Healthy Longevity Study, and provide after the phrase "The study design and sampling method were described previously (8)" some key information with "In brief, ...". These additional details could better explain the reasons of differences between CLHLS and NHANES. Probably, a table with some core characteristics of both surveys would allow to present them in a clear and concise way.

Response: We thank the editor for the suggestion. We created a supplementary table 1 to describe the characteristics of the two datasets in more detail.

3. Clearly indicate in the "Methods" whether the determination of serum creatinine measurement procedures has been standardized to isotope dilution mass spectrometry (IDMS).

Response: Serum creatinine was standardized to IDMS in NHANES. We stated that serum creatinine was determined by the picric acid method in CLHLS in Methods section. We added this as a study limitation that we currently could not remedy right now. 

4. Provide in the "Data sharing statement" information about the data availability (not "additional" but the surveys data).

Response: We thank the editor for this feedback. NHANES and CLHLS have separate data access policies. We included this statement in the bottom of the methods section (Lines 248-253):

The NHANES data that support the findings of this study are openly available at https://wwwn.cdc.gov/nchs/nhanes/continuousnhanes/default.aspx while the CLHLS data are available on request at https://sites.duke.edu/centerforaging/programs/chinese-longitudinal-healthy-longevity-survey-clhls/data-downloads/.

5. Please use "urinary albumin" instead of "urinary microalbumin".

Response: We thank the editor for pointing this out. We changed "urinary microalbumin" to "urinary albumin". 

6. Use IQR instead of range for the description of ACR and other biomarkers in the manuscript.

Response: We thank the editor for this helpful comment. We agree with the reviewers that range might be driven by extreme outlines, and IQR gives the reader a better sense of the data. We revised the description as below (Lines 310-312, 324-325):

“There was a large difference in the urinary microalbumin between China and the US. The urinary microalbumin in China was lower than in the US (mean: 25.0 vs. 76.4, median: 5 vs. 11.2, IQR: 17.8 vs. 11.2 unit: mg/L).

…

The ACR in China was also significantly lower than in the US (mean: 41.7 vs. 85.0, median: 5.8 vs. 11.1, IQR: 20.7 vs. 20.5, unit: mg/g).”

7. Instead of "Predictors of CKD" that could be revealed in the longitudinal study, the cross-sectional part of the current analysis (related to Table 2) should use the terminology "Factors associated with CKD".

Response: We thank the reviewer for this comment. In publications, there is an increasing trend of denying the use of causal language. Our thinking is that if something is a statistical predictor, then we can call it a predictor. We share the sentiment of the reviewer. We have worked through our manuscript to revise the language throughout (Lines 339-372, 441-450, 529). 

“Factors associated with CKD. There were different factors associated with CKD between China and the US. In China, CKD was found more common in participants who were older, female, with higher household income, bad at self-rated, or had physical activities. However, in the US, CKD was found associated with 

…

In previous studies, it was suggested that CKD was associated with increased age, hypertension, diabetes, cardiovascular diseases, and hyperuricemia in both Chinese and the US population (4). 

…

In conclusion, the elderly population in the US have worse CKD-related biomarker levels than in China, and the factors associated with CKD from demo-social to lifestyle factors vary in the countries.”

8. The data provided in the "Figure 1" is widely available, and should be moved to the Supplementary.

Response: We appreciate the comment on Figure 1. We have moved it to the Supplementary and presented it as supplementary table 3.

9. Change legends in figures: instead of "CKD=1 / CKD=0" use "CKD / no CKD", don't use "Strata" in the legend.

Response: We thank the editor for pointing this out. We have revised the legends in the new figures accordingly.

10. As you correctly identified in the "Methods", the KDIGO defines CKD based on eGFR and albuminuria. The current analysis merge persons with either low eGFR or increased albuminuria into a single "CKD" group, and presents the data according to presence/absence of CKD. It would be a great advantage if, in addition to this already available analysis, the authors will present a separate analysis for "normal vs low eGFR" and "normal vs increased albuminuria" groups.

Response: We added Supplementary Table 5 & 6 of risk factors associated with "normal vs low eGFR" and "normal vs increased ACR", also presented the mortality risk of eGFR groups and abnormal increased ACR in Table 4 & 5. We found (line 368-372, 395-403):

“Most risk factors associated with CKD were also associated with abnormal eGFR and ACR when using abnormal eGFR or ACR as the dependent variable separately. Of note, obesity was risk factor for abnormal eGFR but not for CKD or abnormal ACR in NHANES. Older age was associated with CKD and abnormal eGFR but not with abnormal ACR in CLHLS (see Supplementary table 5 & 6).”

“After stratified by the CKD stages, the population with low eGFR had higher odds of death than those with high eGFR in both CLHLS and NHANES (Table 3, Figure 2). After adjusted for all covariates, compared with the group with eGFR≥90 mL/min per 1.73 m2, the HRs (95% CI) of the elderly whose eGFR under 30 remained significant in both CLHLS and NHANES [1.801 (1.054-3.075) in CLHLS, 3.393 (1.634, 7.044) in NHANES], while eGFR in [30,45) did not increase mortality risk significantly in CLHLS [HRs (95% CI): 1.437 (0.903,2.287)], but had doubled the mortality risk compared with those with eGFR≥90 in NHANES [HRs (95% CI): 2.194 (1.113, 4.325)]. Those with abnormal ACR (≥30) had higher mortality risk than those with normal ACR in both CLHLS and NHANES.”

11. Please review the tables' titles for clarity (both in the main manuscript and Supplement).

Response: We edited the table titles to make them clearer:

Table 1-1. Characteristics and mean (SD) of biomarkers of participants (China: CLHLS 2011).

Table 2. Odds ratio (95% CI) of factors associated with CKD in Chinese

Table 4. Hazard ratio (95% CI) of biomarkers on mortality in Chinese

12. Please unify the data representation in tables. In the Supplementary table 2 some cells have no decimal signs, some have (or there is some mistyping): for example, in the ACR column for different age groups the values are "8.4 (5.4,17.9)", "10.3 (6,22.7)", etc.

Response: We unified the decimal signs in tables.

13. Please clarify in "Methods" the phrase "In the logistic regression and the Cox models, missing value of covariates were not excluded, but coded as a categorical variable."

Response: We thank the editor for this comment. We have clarified the phrase as below (Lines 241-242):

“Missing value of covariates were coded as a categorical variable and included in the logistic regression and the Cox models.”

Comments from Reviewers

Reviewer #1: 

1. Summary

The study summarizes chronic kidney disease biomarkers and mortality among elderly participants in the NHANES survey in the United States and the CLHLS survey in China. It reports on many biomarkers, highlighting a few where they identify differences. The article also reports differences in the mortality among CKD patients in the two surveys and briefly hypothesizes some possible reasons for the differences. However, the authors do not discuss the implications of the overall findings (especially where inconsistencies were found) but rather they summarize tabulated data. More importantly, they overstate the generalizability of the results because they did not incorporate the survey weights into their analyses. This last point is the primary weakness of the study and makes it impossible to accurately assess differences between the two populations. I recommend major revisions before acceptance.

Response: We thank the review for the feedback. We added weights in the supplementary analysis, per the editor’s recommendation. Besides, we enhanced the Discussion section by interpreting the inconsistent results (see the response to the fifth comment by the editor). Our implication wording is also revised to reflect the representativeness of our data source. We hope that future cohorts can be used to validate what we found here. 

2. Areas for improvement

Major issues:

The authors should repeat the analysis with inclusion of the survey weights of each survey. This current omission prevents comparison and generalizability to the populations these surveys represent. It was briefly noted in the Discussion (lines 474-477) that weights were not used, with no discussion of why or what the implications are. This was a few sentences after the cohort size and representativeness of the samples was incorrectly touted as a strength of the study (lines 467-477). The conclusion (lines 484-487) generalizes to the countries by stating differences were found between “the elderly population in the US” and “in China”, which the study has failed to show.

Response: We appreciate the helpful comments from the reviewer. 

We added weighted results in supplementary materials and discussed the implication of the use of weight in Discussion (Line 496-504). Please see our response to the first comment of the editor, which covers stratified analysis to explore generalizability concerns. 

We replaced the two countries with the two cohort’s names to interpret the results with more caution. (Line 528-532). However, we see that the vast majority of publications on this topic relies on hospital based sampling, which is less representative than community or population-based cohorts, we do believe this is an advantage. But the reviewer is right, non-standardized recruitment and measurements may pose issues.

Also, as the age distributions of the survey participants are quite different between the two samples (lines 246-247), comparison should occur within detailed age groups, or age-standardized across included age groups. This will more accurately highlight real differences in biomarkers or mortality that are not simply due to ageing. For example, reported prevalence of CKD in line 264 could also be reported as age-standardized to better compare the two samples. The age breakdown in lines 267-270 is helpful and better conveys the differences (higher rates in all age groups in US sample than in China sample) than the overall rates in line 264 (higher overall rate in China sample), which is misleading. This is also an issue in the discussion, where line 410 claims that eGFR is lower in the China sample than in NHANES, but line 413 acknowledges that the age-specific eGFR was higher in the Chinese sample for all but the oldest age group. The authors should take care to report and interpret meaningful results.

Response: We added age group stratified analyses and compared the results within detailed age groups. Please see our response to the second and third comments of the editor. We appreciate the reviewer’s input, which identified key issues that was of concern by other reviewer as well as the editor. Our efforts in doing stratified analyses presented more robust results. 

The conclusion of the abstract supposes that some of the differences may be due to co-morbid conditions (line 53). But this would easily be partially accounted for by at least including indicators for diabetes and hypertension in the adjusted model. Calculating separate results among those with diabetes and/or hypertension is coarse and not very useful, as the authors themselves note that diabetes and hypertension prevalence differs quite a bit between the study populations. Given the known associations between these two conditions and CKD, they should be included in the adjusted models.

Response: We revised the results tables by including diabetes and hypertension in the description table and adjusted model. The association between risk factors (like age, gender, and SES) and CKD were not affected by this adjustment. The association between CKD and mortality were also not affected significantly. Before and after adjusting for hypertension and diabetes, the HRs (95% CI) of CKD on mortality were 1.104 (0.951, 1.280) vs. 1.091 (0.94, 1.266) in CLHLS and 2.067 (1.488, 2.872) vs. 2.179 (1.561, 3.041) in NHANES. However, we admitted that we only adjusted for a limited number of comorbidities in Line 484-487.

In the discussion, differences were repeated but the “why” was not explored. For example, lines 389-394 report that females had higher levels of CKD in the Chinese sample, but there was no association between sex and CKD diagnosis in the US sample. But there was no discussion about why this might be observed. Also, in the US, high household income and low education level were associated with CKD prevalence (lines 392-393) but these two characteristics are typically at odds with each other; usually higher income households have more education. I wish this apparent paradox had been somewhat explored or explained.

Response: We presented some possible reasons to explain the difference mainly in the subsection of the Discussion (Line 466-497). We found a possible reason for the gender difference in CKD between the Chinese sample and the US sample in Line 481-483: “Therefore, we assumed the elder women in China might have a high rate of reproductive tract infection several decades ago, which might have led to CKD later, as poor personal hygiene and low living standard at that time.”

We put both income and education in the model. High household income and low education level were both associated with CKD prevalence, which means despite the education level, high income was still associated with CKD and same for education. From an epidemiologic perspective, a couple of underlying drivers may be at play. There may be healthy worker selection effect (a type of selection bias), confounder by some underlying socioeconomic indicator, or related to occupational determinants of health. Perhaps those with lower education tends to work in jobs that are more detrimental to health, and to achieve higher income it carriers a heavier toll on the body. All of this is speculative, so we did not add our guesses to the discussion section. Our study is not able to identify the underlying causes, however, through an analysis of the literature, we see that those with lower income (couple of education) should have poorer prognostic for CKD as a form of non-communicable disease. It is also that our finding is a statistical aberration.

Minor issues:

GBD 2019 has been published for a while now and should replace the GBD 2017 source.

Response: There was no 2019 paper specifically focused on CKD published like 2017 CKD paper yet. So we updated the data using GBD 2019 data tool from the IHME website:

“In 2019, the estimated age-standardized prevalence of CKD in China and the US were 8125 and 8179 per 100,000, respectively. The percentage of changes in age-standardized prevalence rates between 2010 and 2019 of the two countries were 0.6% and 4% respectively. Age-standardized mortality of CKD in China in 2019 was 11.2 per 100 000 (95% CI: 9.6-12.8), lower than that in the US (17.8 per 100 000 [95% CI: 16.1, 18.9]), and the age-standardized mortality change between 2010 and 2019 were also different (-10.1% [95% CI: -23.1, 3.1] vs. 7.1% [95% CI: 3.4, 11.3])”

What are the implications of the differences in CKD biomarker measurement techniques? This is vaguely included as a limitation (lines 477-478) but not discussed at all in terms of how it affects interpretation of results. The title of the article should be revised to place less emphasis on biomarkers if the authors are not confident about the comparability of biomarker measurement between the samples or able to discuss the potential bias in more detail.

Response: Measurement techniques differences can cause random error bias or systematic bias. For these two studies, we do not know of any proven citations or examples that the two measurement techniques yield different results. However, our analysis is still the evidence for comparison of elderly populations in these two countries. 

Baseline characteristics of % diabetes and % hypertension among participants should be reported.

Response: We agreed and added characteristics of % diabetes and % hypertension in table 1. 

The reported in-text results were cumbersome to read and difficult to draw conclusions from as a reader, but I acknowledge that can be difficult when reporting many different variables. (examples: lines 359-360 in particular, and paragraphs that contain lists of CKD prevalences)

Response: Our writing may be wordy, because we aimed to present the findings completely, however, for better readability, we will work with the editorial staff of the journal to ensure better presentation.

It is unclear why the authors also conducted the analysis using the MDRD equation for the NHANES sample. Written justification would have strengthened the decision to include it.

Response: We did analyses using both MDRD and CKD-EPI. We choose to remove the MDRD for NHANES results since a recent meta-analysis has shown that the performance was more accurate for CKD-EPI creatinine equation than the MDRD equation (PMID: 29046330). 

Positive correlations should be assessed for both statistical significance and clinical relevance. The reported OR of 1.000 (1.000, 1.000) in line 351 is apparently statistically significant (with more decimal places?) but is meaningless clinically.

Response: This is a good point, because PLOS One is a general journals, we will also ensure our results adhere to the guidelines for publication, statistical meaningfulness, and also clinical relevance. At a later stage, we will work with the editorial staff to ensure that the reporting guidelines are met. 

Edit for clarity:

- “bad at self-rated” (line 319)

- “associated with … White, other races…” (line 321)

- “In previous studies…” should be “In a previous study…” (line 416)

Response: We replaced “bad at self-rated” with “self-rated bad health”, “associated with … White, other races…” with “participants with older age, education below high school, self-rated bad health, belonging to white or other races, were not married, or currently smoking were more likely to have CKD”, and “In previous studies…” with “In a previous study…”.

A study cited in the discussion found male sex predictive of CKD in a Chinese sample, which the authors claim “is consistent with our results (line 419). But line 389 states that female sex is predictive of CKD in this study’s Chinese sample, which is the opposite of the cited study.

Response: We thank the reviewer for pointing this out. We corrected “male sex predictive of CKD in a Chinese sample” in the discussion. It should be “being female was associated with decreased eGFR in China but not in the US”, which was consistent with our results. 

I’m curious if the comparison studies described in the Discussion adjusted their models with the same or similar list of variables, or if differences in variables could account for some of the different results. (for example the study in lines 436-437 that reported different results than this study)

Response: Previous studies adjusted for similar covariates as ours: age, sex, ethnic origin, history of cardiovascular disease, blood pressure, diabetes, and smoking. Some studies also adjusted for cholesterol level. We additionally adjusted for household income, alcohol drinking, physical activity, and self-reported health condition. We stated that we adjusted for a limited number of comorbidities (Line 484-489). Meanwhile, we found the adjustment of hypertension and diabetes did not affect the results significantly.

Was prevalence of chronic glomerulonephritis available in either of the surveys? If so, it should have been included in the adjusted models. If not, it would have been helpful to have it stated explicitly (since diabetes and hypertension were available). The one mention of this common primary renal diagnosis, especially in China, was insufficient given its potential importance to some of the differences shown. In renal registries, for example, survival among patients on renal replacement therapy is sometimes reported by primary renal diagnosis (especially diabetes, hypertension, chronic glomerulonephritis, and perhaps other causes), which would have supported the authors’ assertion that underlying comorbidities may account for differences in mortality. I believe from registries, that mortality is higher among patients with ESRD due to diabetes than those with ESRD due to chronic glomerulonephritis. So if more US patients have diabetes than the China samples, and vice versa with chronic glomerulonephritis, then this would be an easy connection to make to support the results.

Response: we did not have available diagnosis data for the chronic glomerulonephritis. We stated it in Line 483-486: “However, we did not examine the comprehensive comorbidity condition including chronic glomerulonephritis among participants, so further studies may need to explore the gap of comorbidities in the Chinese elderly and those in the US.”

More US participants have self-reported diabetes than the China samples in our data. Before and after adjusting for hypertension and diabetes, the HRs (95% CI) of CKD on mortality were 1.104 (0.951, 1.280) vs. 1.091 (0.94, 1.266) in CLHLS and 2.067 (1.488, 2.872) vs. 2.179 (1.561, 3.041) in NHANES. 

I wasn’t clear by the end of the article exactly what the purpose of the study was or what conclusions I should draw from it are. The results can certainly be useful, but I wish the Introduction and Discussion had built up more motivation for and implications of the study.

Figures:

- Figure 1 should be a table.

- Figures 2 and 3 should be clearly labeled with CLHLS and NHANES.

Response: Figure 1 was presented as Supplementary table 3 now. We added CLHLS and NHANES title to the figure 1 and 2 (previous figure 2 & 3).

Reviewer #2:

Thanks for sharing this interesting work. Differences between populations are always a reason for hypothesis for future research regarding their causality (factors, genetic, racial, dietary or others). Very extensive information has been collected and compared, which is difficult to show clearly and is not at times overwhelming and confusing in its presentation.

The study hypothesis is clearly stated, and the population used is adequate.

The methodology is adequate, using population surveys. A problem that will creep into all the results analyzed is the age difference between the two populations. Although multivariate analyzes have been carried out trying to minimize the effect of the age difference between the populations, the use of a matched analysis with more similar populations could have been a useful strategy.

Response: We added age group stratified analyses and compared the results within detailed age groups. Please see our response to the second and third comments of the editor. 

We prefer to use a cohort design to ensure all data is used, however, we agree that using a match analysis will be interesting as well, and perhaps in a later study, we will use hospital nested-case-control study for this.

It looks like a comment at least should be done regarding the BMI classification for older adults (aged 65 or older) since a metanalysis published in 2014 by Winter and Al, has shown a different classification for this population (Winter JE, MacInnis RJ, Wattanapenpaiboon N, Nowson CA. BMI and all-cause mortality in older adults: a meta-analysis. Am J Clin Nutr. 2014 Apr;99(4):875-90) suggesting that optimal BMI for those >65 years is likely between 23-30, instead of 18.5 to 25.

Response: We thank the reviewer for telling us the different optimal BMI for older adults. Based on the general classification, we found “normal (18.5-24.9)” and “Overweight (25.0-29.9)” both had lower odds of CKD compared to “Underweight (<18.5)” in CLHLS. But we found no significant difference when BMI classified as “<23”, “23-30”, and “>30”. The two kinds of BMI classification were both not detected to be associated with CKD in NHANES.

Therefore, we decided to keep the previous more detailed classification.

Regarding the Results

The two populations studied differ widely in the age of the population, with the Chinese population averaging 13 years older (85 vs. 73 years).

This difference can explain many of the other differences found, even some that seem obvious like practicing physical activity. The fact that the Chinese population is much older than the US population should be discussed, it could be a survivor bias, being a "healthier" population.

Response: We agree with the reviewer’s insights. We added age groups stratified results. Some associations only existed in participants aged 80 or older in CLHLS. We also talked the age difference in Line 435-437:

“Although the mean eGFR in our study was lower in China, it may result from the larger proportion of participants aged 80 and above in CLHLS than NHANES. The age-specific eGFR was higher in the Chinese participants, except for the eldest group.”

 We further discussed about possible survivor bias in the Line 494-497:

“In addition, the difference of the findings between the two countries varied among different age groups. There were a much larger proportion of the population aged 80 or older in the China sample than the US sample (65.5% vs. 26.1%). There may be survivor bias caused by a relatively healthier Chinese population.”

In relation to the way of presenting the results, which involve a wealth of information, their presentation would be clearer and easier to read if additional headings were incorporated for the different results to be described., As well as for including headings at the discussion.

Response: We agreed with the reviewer’s suggestion. We had subheadings for the Results section and also added subheadings for the Discussion section.

A minor commentary: the subtitle of the prevalence chapter would be more adequately described as “Difference in CKD prevalence between the two countries” or “Prevalence of CKD in both countries”.

Response: We agreed “Prevalence of CKD in both countries” should be more accurate and revised it accordingly.

An additional issue for the mortality analysis may be the loss to follow-up of 321 CLHLS participants (page 13, line 128, representing 14% of the population. It should be investigated whether these participants were not lost to follow-up due to death, or comment that this was ruled out.

Response: CLHLS has a proportion of lost of follow up. The previous CLHLS data quality evaluation study suggested the main reasons for the loss to follow-up were changes in home addresses and reluctance to participate due to transportation difficulties and unfavorable weather [Gu, 2008]. We could not rule out the death for the reason. So we compared the characteristics between those lost of follow-up and those followed up. Those lost in the first follow-up were more likely to have formal education and higher household income, less likely to do physical activity currently than those included in the follow-up study. Meanwhile, they had similar age, gender, race, marriage, smoking, alcohol drinking and BMI distribution. We added these observations to the methods section.

Regarding the limitations of the study, it should be mentioned the differences in the age of both population, that could itself explain many differences between both populations.

Response: we agreed to this and added this limitation in the discussion. (Line 521-525)

---

## [Decision Letter · Decision Letter 1]

23 Aug 2021

PONE-D-21-07064R1

A Comparison Study of Chronic Kidney Disease Biomarkers and Mortality among Older Adults in China and the United States

PLOS ONE

Dear Dr. Ji,

Thank you for submitting your manuscript to PLOS ONE. After careful consideration, we feel that it has merit but does not fully meet PLOS ONE’s publication criteria as it currently stands. Therefore, we invite you to submit a revised version of the manuscript that addresses the points raised during the review process.

In addition to the Reviewers' comments, please consider the following items:

1. Regarding the extended tables, please consider to use bold style to underline the statistically significant results (this is just a suggestion, not obligatory requirement).

2. Please use 2 decimal signs in OR, i.e. instead of "1.801 (1.054-3.075)" use "1.80 (1.05-3.07)"; except of the cases where more decimal signs are required to demonstrate the 95%CI close to 1 (like "1.0003-1.18").

3. Please revise the manuscript for style describing intervals, and change it to more easier to readers (i.e. avoid technical expressions like "eGFR in [30,45) did not increase mortality risk", and use instead "eGFR 30-45 ml/min/1.73m2")

4. Considering the reviewer's comment "Positive correlations should be assessed for both statistical significance and clinical relevance. The reported OR of 1.000 (1.000, 1.000) in line 351 is apparently statistically significant (with more decimal places?) but is meaningless clinically." and your response "...At a later stage, we will work with the editorial staff to ensure that the reporting guidelines are met....." - Please consider that once the manuscript is accepted for the publication, there will be only minimal grammar/style changes implemented by the Editorial office. Thus, all issues should be resolved during the revision, and this particular reviewer's comment should be addressed properly now, before the "Accepted" decision.

5. The Associate Editor reviewed your manuscript and has indicated "Please assess the reporting of this observational study of clinical data using the STROBE checklist (http://www.strobe-statement.org)"

6. Please use in the "Article summary" the phrases that a reader could understand without reading the manuscript. For example, the phrase "CKD is associated with higher mortality in US study participants, perhaps due to comorbid conditions." would be more clear if changed to something like "CKD is associated with higher mortality in US but not in China study participants, perhaps due to comorbid conditions." Instead of general statement "US study participants had higher concentrations of CKD-related biomarker levels compared with Chinese participants" use the complete list of biomarkers.

The item "Strengths and limitations of this study. Our data was derived from large nationally-

61 representative cohorts in both countries. However, there are challenges in age

62 distribution differences, standardization of CKD biomarker measurement techniques, as

63 well as covariates definitions." should be rephrased for clarity and conciseness.

7. The description of two groups comparison like "The urinary

312 albumin in China was lower than in the US (mean: 25.0 vs. 76.4, median: 5 vs. 11.2, IQR:

313 17.8 vs. 11.2 unit: mg/L)" or "people with lower education (27.2 mg/L of “no formal education” vs 103.9 mg/L of “below high school”)" has limited meaning. Please use plain language and refer to the table with numeric results, or indicate widely accepted measures of central tendency and variability in the text. This also concerns the description at lines 313-339.

8. Some references have no page numbers or eNumber, or ISBN (where appropriate) - like ref 22, 26 and many others. Please check all references and format appropriately.

9. It seems the Table 2 title should be "Odds ratio (95% CI) of factors associated with CKD in CLHLS 2011 participants (China)"

10. In the Supplementary Table 1. Cohorts’ characteristics of this study it is indicated "Older men and women aged 65 or above" for both cohorts. However, the NHANES includes also persons 18-64 years old. Please could you explain?

We look forward to receiving your revised manuscript.

Kind regards,

Boris Bikbov

Academic Editor

PLOS ONE

Journal Requirements:

Additional Editor Comments (if provided):

Reviewers' comments:

Reviewer's Responses to Questions

**Comments to the Author**

1. If the authors have adequately addressed your comments raised in a previous round of review and you feel that this manuscript is now acceptable for publication, you may indicate that here to bypass the “Comments to the Author” section, enter your conflict of interest statement in the “Confidential to Editor” section, and submit your "Accept" recommendation.

Reviewer #1: All comments have been addressed

Reviewer #2: All comments have been addressed

2. Is the manuscript technically sound, and do the data support the conclusions?

Reviewer #1: Yes

Reviewer #2: Partly

3. Has the statistical analysis been performed appropriately and rigorously? 

Reviewer #1: Yes

Reviewer #2: Yes

4. Have the authors made all data underlying the findings in their manuscript fully available?

Reviewer #1: Yes

Reviewer #2: Yes

5. Is the manuscript presented in an intelligible fashion and written in standard English?

Reviewer #1: Yes

Reviewer #2: Yes

6. Review Comments to the Author

Reviewer #1: The authors' changes have improved the manuscript a great deal. Some light copy-editing is needed. One last change I would recommend is including the limitation of much higher loss to follow-up in the CLHLS sample versus the NHANES sample.

Reviewer #2: As a consequence of correcting the text to the recommendations, the article has greatly improved.

Some minor elements persist, some already mentioned previously.

1. In the Methods chapter, on page 9 when describing "Measurement of biomarkers. In CLHLS, the urine was tested for microalbumin" should refer to albumin and not to microalbumin.

2. The diagnosis of hypertension should include, in addition to the presence of BP <140/90, the intake of antihypertensive medication

3. When describing the prevalence of CKD in both populations, some findings are contrary to the usual occurrence, so must be made some comment. In Chinese participants, those who never smoked and did physical activity, a higher prevalence of CKD was found, this finding should be commented, which is surely more in relation to age than to these characteristics, and is contrary to the effects of smoking and physical activity in general.

7. PLOS authors have the option to publish the peer review history of their article (what does this mean?). If published, this will include your full peer review and any attached files.

Reviewer #1: **Yes: **Sarah Wulf Hanson

Reviewer #2: **Yes: **Laura Sola

---

## [Author Response · Author response to Decision Letter 1]

8 Sep 2021

Reviewer #1: The authors' changes have improved the manuscript a great deal. Some light copy-editing is needed. One last change I would recommend is including the limitation of much higher loss to follow-up in the CLHLS sample versus the NHANES sample.

Response: We thank the reviewer for this feedback. We added this point at the end of the Discussion (Line 621-625):

“Last but not least, there were more participants lost to follow-up in CLHLS than in NHANES. Those lost to follow-up in CLHLS had similar characteristics with those did not lost to follow-up except that they were more likely to have formal education and higher household income. This can make the mortality analyses results of two countries less comparable to a small extent.”

Reviewer #2: As a consequence of correcting the text to the recommendations, the article has greatly improved.

Some minor elements persist, some already mentioned previously.

1. In the Methods chapter, on page 9 when describing "Measurement of biomarkers. In CLHLS, the urine was tested for microalbumin" should refer to albumin and not to microalbumin.

2. The diagnosis of hypertension should include, in addition to the presence of BP <140/90, the intake of antihypertensive medication

3. When describing the prevalence of CKD in both populations, some findings are contrary to the usual occurrence, so must be made some comment. In Chinese participants, those who never smoked and did physical activity, a higher prevalence of CKD was found, this finding should be commented, which is surely more in relation to age than to these characteristics, and is contrary to the effects of smoking and physical activity in general.

Response: We thank the reviewer for giving these helpful comments. We have revised them accordingly.

First, we double-checked the definition of microalbumin and albumin, and then changed the item “microalbumin” into “albumin”. 

Second, we presumed that the comparability of hypertension-related results is more important than the accuracy of hypertension definition in this comparative study. Since there was no medication data in CLHLS, we used the current definition of hypertension so that it was applicable for all participants in both countries. 

Third, we explained this counterintuitive result in Line 538-543:

“Furthermore, contrary to the positive health effect of physical activity in general, we found no physical activity was associated to lower risk of having CKD among Chinese participants. We speculated that the presence of CKD might be more related to age than to physical activity. It is possible that doing exercise enable people to live longer but, thus, have CKD. This may also explain why never smokers among Chinese elderly showed a higher prevalence of CKD.”

---

## [Decision Letter · Decision Letter 2]

7 Oct 2021

PONE-D-21-07064R2A Comparison Study of Chronic Kidney Disease Biomarkers and Mortality among Older Adults in China and the United StatesPLOS ONE

Dear Dr. Ji,

Thank you for submitting your manuscript to PLOS ONE. After careful consideration, we feel that it has merit but does not fully meet PLOS ONE’s publication criteria as it currently stands. Therefore, we invite you to submit a revised version of the manuscript that addresses the points raised during the review process.

We look forward to receiving your revised manuscript.

Kind regards,

Boris Bikbov

Academic Editor

PLOS ONE

Journal Requirements:

Reviewers' comments:

Reviewer's Responses to Questions

**Comments to the Author**

1. If the authors have adequately addressed your comments raised in a previous round of review and you feel that this manuscript is now acceptable for publication, you may indicate that here to bypass the “Comments to the Author” section, enter your conflict of interest statement in the “Confidential to Editor” section, and submit your "Accept" recommendation.

Reviewer #1: (No Response)

Reviewer #2: All comments have been addressed

2. Is the manuscript technically sound, and do the data support the conclusions?

Reviewer #1: Yes

Reviewer #2: Yes

3. Has the statistical analysis been performed appropriately and rigorously? 

Reviewer #1: Yes

Reviewer #2: Yes

4. Have the authors made all data underlying the findings in their manuscript fully available?

Reviewer #1: Yes

Reviewer #2: Yes

5. Is the manuscript presented in an intelligible fashion and written in standard English?

Reviewer #1: Yes

Reviewer #2: Yes

6. Review Comments to the Author

Reviewer #1: This manuscript has improved, but a few remaining issues need to be resolved.

1. In the Abstract conclusion, you state that "the association between CKD and mortality was also stronger among the US older adults. This may be due to age difference, biological differences, or co-morbid conditions." However, the statistical models control for age, and so the estimated hazard ratios control for age. So age differences would then not contribute to the difference in strength of association. I suggest removing "age difference" from the last sentence.

2. The last sentence of the Introduction seems states a result rather than a hypothesis. Restate as a hypothesis.

3. Check all references for accuracy (line 94 says "refs", for example, rather than reference numbers).

4. Line 243 states that "age was adjusted for as categorical in the logistic models" but does not state how. Better would be "age was included as four separate age bins (65-69, 70-74, 75-79, 80+) in the logistic models".

5. Line 518-526 discusses the rationale for excluding sample weights from the current analysis. I find this reasoning unconvincing, especially because CLHLS oversampled rural areas and NHANES minorities and lower SES, and the non-weighted results are therefore clearly non-representative. The analysis including sample weights should be in the main text, especially as NHANES has not issued guidance to ignore sample weights in that survey.

Reviewer #2: The questions posed have been answered in an acceptable way. In its new edition the article has been vastly improved, and is fit to be published.

7. PLOS authors have the option to publish the peer review history of their article (what does this mean?). If published, this will include your full peer review and any attached files.

Reviewer #1: **Yes: **Sarah Wulf Hanson

Reviewer #2: **Yes: **Laura Sola

---

## [Author Response · Author response to Decision Letter 2]

8 Oct 2021

Dear Editor,

Thank you for the opportunity for an additional round of revision on our manuscript. Our point-by-point response are below.

Reviewer #1: This manuscript has improved, but a few remaining issues need to be resolved.

1. In the Abstract conclusion, you state that "the association between CKD and mortality was also stronger among the US older adults. This may be due to age difference, biological differences, or co-morbid conditions." However, the statistical models control for age, and so the estimated hazard ratios control for age. So age differences would then not contribute to the difference in strength of association. I suggest removing "age difference" from the last sentence.

Response: We thank the reviewer for pointing this out. We removed “age difference” from the last sentence of the abstract (Line 53-54):

“This may be due to the biological differences, or co-morbid conditions.”

2. The last sentence of the Introduction seems states a result rather than a hypothesis. Restate as a hypothesis.

Response: We thank the reviewer for this feedback. We revised the sentence as below (Line 101-103):

“In addition, the association between CKD and mortality might also be different among the US elderly population and the Chinese.”

3. Check all references for accuracy (line 94 says "refs", for example, rather than reference numbers).

Response: We thank the reviewer for the reminding. We found Line 94-95 did not need a reference, since it referred to the articles mentioned above. Therefore, we removed “(refs)”, and revised the sentence as below. We also checked all references carefully.

“However, these previous studies mainly focused on comparing the prevalence of CKD in the general population between China and the US, the differences in the prevalence of CKD among elderly populations between China and the US, and their association with mortality was not known yet.”

4. Line 243 states that "age was adjusted for as categorical in the logistic models" but does not state how. Better would be "age was included as four separate age bins (65-69, 70-74, 75-79, 80+) in the logistic models".

Response: We appreciate the reviewer for this helpful comment. We have rephrased as below in Line 248-249:

“Besides, age was adjusted for as four separate age bins (65-69, 70-74, 75-79, 80+) in the logistic models and as continuous in the Cox models.”

5. Line 518-526 discusses the rationale for excluding sample weights from the current analysis. I find this reasoning unconvincing, especially because CLHLS oversampled rural areas and NHANES minorities and lower SES, and the non-weighted results are therefore clearly non-representative. The analysis including sample weights should be in the main text, especially as NHANES has not issued guidance to ignore sample weights in that survey.

Response: We concur with the reviewer that weighting the sample would allow us to make generalizable statements about the population at large. However, we prefer not to include the weights in the main text for reasons stated in the last revision, plus we are not making population at large statements. Treating each participant as an individual allows us to avoid some bias that may be introduced by weights, and since the motivation of our study is to look at epidemiological dose-response relationship, rather make population prevalence estimates, our method is perhaps more robust statistically. Furthermore, we presented both weighted and unweighted results for the reader. Once again, we concur with the reviewer, but prefer to exercise our discretion as authors in making this decision. If necessary, we hope the editor can make a recommendation as well.

Reviewer #2: The questions posed have been answered in an acceptable way. In its new edition the article has been vastly improved, and is fit to be published.

Response: We the reviewer for the feedback.

---

## [Decision Letter · Decision Letter 3]

14 Oct 2021

PONE-D-21-07064R3A Comparison Study of Chronic Kidney Disease Biomarkers and Mortality among Older Adults in China and the United StatesPLOS ONE

Dear Dr. Ji,

Thank you for submitting your manuscript to PLOS ONE. After careful consideration, we feel that it has merit but does not fully meet PLOS ONE’s publication criteria as it currently stands. Therefore, we invite you to submit a revised version of the manuscript that addresses the points raised during the review process.

We look forward to receiving your revised manuscript.

Kind regards,

Boris Bikbov

Academic Editor

PLOS ONE

Journal Requirements:

Reviewers' comments:

Reviewer's Responses to Questions

**Comments to the Author**

1. If the authors have adequately addressed your comments raised in a previous round of review and you feel that this manuscript is now acceptable for publication, you may indicate that here to bypass the “Comments to the Author” section, enter your conflict of interest statement in the “Confidential to Editor” section, and submit your "Accept" recommendation.

Reviewer #1: (No Response)

2. Is the manuscript technically sound, and do the data support the conclusions?

Reviewer #1: Yes

3. Has the statistical analysis been performed appropriately and rigorously? 

Reviewer #1: Yes

4. Have the authors made all data underlying the findings in their manuscript fully available?

Reviewer #1: Yes

5. Is the manuscript presented in an intelligible fashion and written in standard English?

Reviewer #1: Yes

6. Review Comments to the Author

Reviewer #1: I thank the authors for their quick response. I accept that they will not include sample weights in the main analysis of the paper and am glad the Supplement includes them, and I now suggest two remaining changes:

1. Change statements that assert national representativeness

In the authors' last response, they state "However, we prefer not to include the weights in the main text for reasons stated in the last revision, plus we are not making population at large statements. Treating each participant as an individual allows us to avoid some bias that may be introduced by weights, and since the motivation of our study is to look at epidemiological dose-response relationship, rather make population prevalence estimates, our method is perhaps more robust statistically." In the Results section, the authors are careful to restrict statements to the study samples, but there remain many sentences that do still refer to "the US" or "China" (or similar generalized references), and there are many sentences that do make population prevalence estimates, contrary to the authors' response. Before publication, results need to be changed to refer to the study samples (such as "CLHLS sample" or "US participants", etc) rather than the countries in all of the following lines:

282-285 (example wording change included below), 287, 303, 305, 309, 311, 312, 313, 315-326 ("both countries" change to "both samples"), 332 ("two countries" change to "both samples"), 343, 362, 421, 423, 442, 450, 456.

Original text for lines 282-285: "Among people over 65 years of age, the prevalence of CKD in China and the US were 44.4% (95% CI: 42.2%-46.6%) and 42.3% (95% CI: 40.2%-44.6%), respectively. Besides, the CKD prevalence for the participants without diabetes and hypertension were 37.8% in China and 33% in the US (diabetes/hypertension prevalence: 58% in China vs. 50% in the US)."

Suggested change: "Among people over 65 years of age, the prevalence of CKD in the CLHLS and the NHANES samples were 44.4% (95% CI: 42.2%-46.6%) and 42.3% (95% CI: 40.2%-44.6%), respectively. Besides, the CKD prevalence for the participants without diabetes and hypertension were 37.8% in CLHLS and 33% in NHANES (diabetes/hypertension prevalence: 58% in CLHLS vs. 50% in NHANES)."

2. Clarify that the sample weight limitation applies to both surveys.

Lines 525-526 original text: "We only presented the results considering sample weight as supplementary materials."

Suggested change: "We excluded sample weights from CLHLS for this reason and from NHANES for consistency, and we present results incorporating the sample weights for both studies in the supplementary materials."

I thank the authors for the opportunity to provide feedback on this manuscript, and I look forward to seeing it published soon.

7. PLOS authors have the option to publish the peer review history of their article (what does this mean?). If published, this will include your full peer review and any attached files.

Reviewer #1: **Yes: **Sarah Wulf Hanson

---

## [Author Response · Author response to Decision Letter 3]

15 Oct 2021

Dear Editor,

Thank you for the opportunity for an additional round of revision on our manuscript. The reviewer was very fast in providing additional feedback, we appreciate this. To address the final edit, our point-by-point response are below.

Reviewer #1: I thank the authors for their quick response. I accept that they will not include sample weights in the main analysis of the paper and am glad the Supplement includes them, and I now suggest two remaining changes:

1. Change statements that assert national representativeness

In the authors' last response, they state "However, we prefer not to include the weights in the main text for reasons stated in the last revision, plus we are not making population at large statements. Treating each participant as an individual allows us to avoid some bias that may be introduced by weights, and since the motivation of our study is to look at epidemiological dose-response relationship, rather make population prevalence estimates, our method is perhaps more robust statistically." In the Results section, the authors are careful to restrict statements to the study samples, but there remain many sentences that do still refer to "the US" or "China" (or similar generalized references), and there are many sentences that do make population prevalence estimates, contrary to the authors' response. Before publication, results need to be changed to refer to the study samples (such as "CLHLS sample" or "US participants", etc) rather than the countries in all of the following lines:

282-285 (example wording change included below), 287, 303, 305, 309, 311, 312, 313, 315-326 ("both countries" change to "both samples"), 332 ("two countries" change to "both samples"), 343, 362, 421, 423, 442, 450, 456.

Original text for lines 282-285: "Among people over 65 years of age, the prevalence of CKD in China and the US were 44.4% (95% CI: 42.2%-46.6%) and 42.3% (95% CI: 40.2%-44.6%), respectively. Besides, the CKD prevalence for the participants without diabetes and hypertension were 37.8% in China and 33% in the US (diabetes/hypertension prevalence: 58% in China vs. 50% in the US)."

Suggested change: "Among people over 65 years of age, the prevalence of CKD in the CLHLS and the NHANES samples were 44.4% (95% CI: 42.2%-46.6%) and 42.3% (95% CI: 40.2%-44.6%), respectively. Besides, the CKD prevalence for the participants without diabetes and hypertension were 37.8% in CLHLS and 33% in NHANES (diabetes/hypertension prevalence: 58% in CLHLS vs. 50% in NHANES)."

Response: We concur, and revised the wording as below:

Title has been revised to: 

“Title: Chronic Kidney Disease Biomarkers and Mortality among Older Adults: A Comparison Study of Survey Samples in China and the United States”

Line 282-286:

“Among people over 65 years of age, the prevalence of CKD in CLHLS and NHANES samples were 44.4% (95% CI: 42.2%-46.6%) and 42.3% (95% CI: 40.2%-44.6%), respectively. Besides, the CKD prevalence for the participants without diabetes and hypertension were 37.8% in CLHLS and 33% in NHANES (diabetes/hypertension prevalence: 58% in CLHLS vs. 50% in NHANES).”

Line 286-287:

“However, the prevalence in each age group of NHANES participants was higher than Chinese participants, …”

Line 304:

“… were lower in the Chinese participants than the US sample …”

Line 306:

“… while the BUN (mmol/L) was higher in the Chinese sample (6.9 vs 6.1) (Table 1).”

Line 309-315:

“The mean difference between the CLHLS and NHANES varied in different age groups. The urinary creatinine, BUN and eGFR were both higher in CLHLS participants than NHANES sample aged younger than 80 and this difference almost disappeared in participants aged 80 and older. The eGFR decreased with age increasing in both CLHLS and NHANES samples. The urinary albumin, ACR, serum creatinine, and uric acid were lower in Chinese participants than the US ones among different age groups consistently. Plasma albumin level was similar in both samples across all age groups.”

Line 317-328:

“There was a large difference in the urinary albumin between the two samples. The mean of urinary albumin level of CLHLS participants was 25.0 mg/L (SD: 75.4), much lower than that of NHANES participants (76.4 mg/L, SD: 359.6). Among Chinese participants, higher urinary albumin level usually appeared along with factors like age over 80 (29.5 mg/L), female (25.8 mg/L), high household income (29.8 mg/L), widowed (28.6 mg/L), good health condition (25.7 mg/L), never smoker (26.6 mg/L), and former drinker (34.7 mg/L) (Table 1-1). However, among NHANES participants, people had higher urinary albumin were more likely to be aged 75-79 (109.3 mg/L), male (98.1 mg/L), with low household income (88.8 mg/L), separated (134.2 mg/L), with poor health condition (171.8 mg/L), current smoker (111.6 mg/L), and never drinker (98.0 mg/L) (Table 1-2). Besides, some characteristics were found related to higher urinary albumin level in both samples, including lower education, obese, insufficient physical activity, hypertension, and diabetes (Table 1-1 & 1-2).”

Line 332-334:

“However, we did not find a similar pattern in the two samples in terms of the relationship between ACR and other demographic or lifestyle characteristics.”

Line 345:

“There were different factors associated with CKD between CLHLS and NHANES samples.”

Line 363-365:

“Among NHANES participants, current smokers also showed about 40% greater odds of having CKD. Besides, in the Chinese sample, those who had no physical activities had lower odds of CKD (0.708, 95% CI: 0.534, 0.938) than the others.”

Line 424-427:

“In the US sample, gender was not associated with CKD, while high household income and low education level were associated with a higher prevalence of CKD. Furthermore, the association between CKD/eGFR and mortality was found stronger in the US sample.”

Line 445:

“Although the mean eGFR in our study was lower in in the Chinese sample, …”

Line 452-453:

“This study also suggested that being female was associated with decreased eGFR in CLHLS participants but not in the NHANES sample, …”

Line 458-460:

“The weaker relationship between higher BMI and CKD may be reasonable considering the huge gap between the proportion of overweight people in the Chinese and the US samples.”

2. Clarify that the sample weight limitation applies to both surveys.

Lines 525-526 original text: "We only presented the results considering sample weight as supplementary materials."

Suggested change: "We excluded sample weights from CLHLS for this reason and from NHANES for consistency, and we present results incorporating the sample weights for both studies in the supplementary materials."

I thank the authors for the opportunity to provide feedback on this manuscript, and I look forward to seeing it published soon.

Response: We revised the sentence as suggested (Line 529-531):

“We excluded sample weights from CLHLS for this reason and from NHANES for consistency, and we present results incorporating the sample weights for both studies in the supplementary materials.”

---

## [Decision Letter · Decision Letter 4]

3 Nov 2021

Chronic Kidney Disease Biomarkers and Mortality among Older Adults: A Comparison Study of Survey Samples in China and the United States

PONE-D-21-07064R4

Dear Dr. Ji,

We’re pleased to inform you that your manuscript has been judged scientifically suitable for publication and will be formally accepted for publication once it meets all outstanding technical requirements.

Kind regards,

Boris Bikbov

Academic Editor

PLOS ONE

Additional Editor Comments (optional):

Reviewers' comments:

Reviewer's Responses to Questions

**Comments to the Author**

1. If the authors have adequately addressed your comments raised in a previous round of review and you feel that this manuscript is now acceptable for publication, you may indicate that here to bypass the “Comments to the Author” section, enter your conflict of interest statement in the “Confidential to Editor” section, and submit your "Accept" recommendation.

Reviewer #1: All comments have been addressed

2. Is the manuscript technically sound, and do the data support the conclusions?

Reviewer #1: Yes

3. Has the statistical analysis been performed appropriately and rigorously? 

Reviewer #1: Yes

4. Have the authors made all data underlying the findings in their manuscript fully available?

Reviewer #1: Yes

5. Is the manuscript presented in an intelligible fashion and written in standard English?

Reviewer #1: Yes

6. Review Comments to the Author

Reviewer #1: I thank the authors for their final changes and for being willing to go through multiple rounds of revisions. I think this manuscript is now ready for publication.

7. PLOS authors have the option to publish the peer review history of their article (what does this mean?). If published, this will include your full peer review and any attached files.

Reviewer #1: **Yes: **Sarah Wulf Hanson

---

## [Editor Report · Acceptance letter]

3 Jan 2022

PONE-D-21-07064R4 

Chronic Kidney Disease Biomarkers and Mortality among Older Adults: A Comparison Study of Survey Samples in China and the United States 

Dear Dr. Ji:

I'm pleased to inform you that your manuscript has been deemed suitable for publication in PLOS ONE. Congratulations! Your manuscript is now with our production department. 

Kind regards, 

on behalf of

Dr. Boris Bikbov 

Academic Editor

PLOS ONE